



# Quantifying agricultural N$_2$O and CH$_4$ emissions in the Netherlands using an airborne eddy covariance system

Paul Waldmann[1], Max Eckl[former 1, now at 9], Leon Knez[1], Klaus-Dirk Gottschaldt[1], Alina Fiehn[1], Christian Mallaun[2], Michał Gałkowski[3,4], Christoph Kiemle[1], Ronald Hutjes[5], Thomas Röckmann[6], Huilin Chen[7,8], and Anke Roiger[1]

[1]Deutsches Zentrum für Luft- und Raumfahrt (DLR), Institut für Physik der Atmosphäre, Oberpfaffenhofen, Germany
[2]Deutsches Zentrum für Luft- und Raumfahrt (DLR), Flugexperimente, Oberpfaffenhofen, Germany
[3]Department of Biogeochemical Signals, Max Planck Institute for Biogechemistry, Jena, Germay
[4]Faculty of Physics and Applied Computer Science, AGH University of Kraków, Kraków, Poland
[5]Earth Systems and Global Change Group, Wageningen University and Research, Wageningen, The Netherlands
[6]Institute for Marine and Atmospheric Research Utrecht, Utrecht University, Utrecht, The Netherlands
[7]Centre for Isotope Research (CIO), University of Groningen, Groningen, The Netherlands
[8]School of Atmospheric Sciences, Nanjing University, Nanjing, China
[9]E.ON Energy Markets GmbH, Essen, Germany

**Correspondence:** Paul Waldmann (Paul.Waldmann@dlr.de)

**Abstract.**

This study reports on the first successful deployment of a new airborne eddy covariance (EC) setup to better characterize and quantify non-CO$_2$ greenhouse gas emissions from agriculture. The system was deployed aboard the DLR research aircraft Cessna Caravan to quantify growing-season emissions of methane (CH$_4$) and nitrous oxide (N$_2$O) in Friesland, an agricultural region in the Netherlands, in early summer 2023. The EC system consists of a commercial quantum cascade laser spectrometer, specifically adapted for airborne observations and providing 10 Hz data of N$_2$O and CH$_4$, and the meteorological measurement suite METPOD, delivering data of the vertical wind, horizontal winds, water vapor and temperature. Our measurements are a novelty for N$_2$O, since they are the first implementation of quantifying agricultural emissions with airborne EC, combining the advantages of regional-scale coverage, while maintaining high spatial resolution and hence are well suited to capture the spatial complexity of this dominant emission sector. The system provides fluxes with minimal low- and high-frequency distortions, low detection limits, and total uncertainties ($30-100\,\%$) comparable to other airborne methods, despite the complexity of agricultural emissions. During measurements in Friesland, we identified clear N$_2$O emission hotspots and hot-moments, with peak fluxes of $0.34\,\mu g\,m^{-2}\,s^{-1}$ on the regional-scale after intensive precipitation following a relatively dry period. Single small-scale hotspot emissions were as high as $1\,\mu g\,m^{-2}\,s^{-1}$. In contrast, CH$_4$ fluxes showed less temporal variations around a mean flux of $1.62\,\mu g\,m^{-2}\,s^{-1}$ throughout the three-week campaign. N$_2$O emissions were relatively high compared to other agricultural regions worldwide, and preliminary comparisons with EDGAR v8.0 and the Dutch emission inventory Emissieregistratie suggest substantial underestimation of growing-season N$_2$O emissions in current inventories and the lack of an appropriate annual cycle. Our results further document the urgent need for independent verification of reported N$_2$O and CH$_4$ emissions



from agriculture, which is the most dominant anthropogenic sector of non-$CO_2$ greenhouse gas emissions and is expected to
become even more dominant in the future, with an increasing world population and food demand.

## 1   Introduction

Methane ($CH_4$) and nitrous oxide ($N_2O$) are the second and third most significant anthropogenic greenhouse gases (GHG)
after carbon dioxide ($CO_2$). Since preindustrial times, atmospheric $CH_4$ concentrations have increased by more than a factor
of 2.5, primarily due to human activities such as fossil fuel use, agriculture, and waste management (Saunois et al., 2024;
Etheridge et al., 1998; Craig et al., 1988). $CH_4$ has a 100-year global warming potential ($GWP_{100}$) of 32 and a relatively short
atmospheric lifetime of 9.1 years, making it a prime target for near-term climate mitigation efforts (IPCC, 2023; Etminan et al.,
2016; Prather et al., 2012). Despite this, $CH_4$ has exhibited record-high atmospheric growth rates in recent years, particularly
in $2020-2022$, after a relatively stable period from $2000-2007$. The reasons for the large increase are not fully understood
(Michel et al., 2024; Thoning et al., 2022; Zhang et al., 2022; Turner et al., 2019; Schaefer et al., 2016; Nisbet et al., 2016).
Observations show that global mean surface $CH_4$ mole fractions follow the shared socio-economic pathway with very high
emissions (SSP 8.5), suggesting that $CH_4$ emissions could seriously undermine sustainable climate targets (Nisbet et al., 2025;
Meinshausen et al., 2020; Nisbet et al., 2019).

$N_2O$, the main precursor of ozone-depleting substances in the stratosphere, has an atmospheric lifetime of about 116 years
and thus contributes to long-term climate change (WMO, 2022; Prather et al., 2015). Its $GWP_{100}$ of 273 is roughly ten times
larger than that of $CH_4$. Measured atmospheric mole fractions already exceed values projected in the highest-emission SSP
scenario, which would lead to a warming of $3.3-5.7°C$ by the end of this century (Tian et al., 2024; IPCC, 2023). This rise
is almost entirely driven by human activities, dominated by the agricultural application of fertilizer, a source that is expected
to increase in the future due to a larger food demand (Tian et al., 2024). $N_2O$ in general will become more relevant in a
decarbonized future, as its concentrations will not decrease until the end of this century, even in the most optimistic SSP
scenario (Meinshausen et al., 2020).

The Paris Agreement aims to limit the global increase in surface temperature to well below $2°C$, better below $1.5°C$ by the
end of the century, compared to the preindustrial level (UNFCCC, 2016). Achieving this goal requires substantial reductions in
GHG emissions and effective mitigation strategies. Although $CO_2$ is the main driver of global warming (IPCC, 2023), cutting
non-$CO_2$ GHGs, especially $CH_4$ and $N_2O$, is also inevitable (Rogelj and Lamboll, 2024; Kanter et al., 2020; Nisbet et al.,
2019). To meet the $1.5°C$ goal, human-made $CH_4$ emissions must decrease by 51 % until 2050 compared to 2020 levels, while
$N_2O$ emissions need to be reduced by 22 % (Rogelj and Lamboll, 2024).

Profound knowledge of GHG emissions, including quantifying sector specific contributions and understanding underlying
processes, is a crucial prerequisite for effective mitigation. Emissions typically are estimated based on bottom-up (BU) methods
using emission fluxes from individual source measurements or from process-based emission models, which are then scaled up
using statistical data. However, BU methods are sometimes incomplete and inaccurate and need to be verified using the so-
called top-down (TD) approach (Nisbet and Weiss, 2010). TD methods rely on atmospheric observations on different scales,





and attribute emissions to specific regions or sectors using measurements of additional tracers, forward transport or inverse modeling approaches. TD methods are furthermore essential to identify possible mitigation targets by providing observational evidence. Until now, most regional to global scale TD measurements have focused on emissions from the fossil fuel sector and

waste management (Förster et al., 2025; Pühl et al., 2024; Krautwurst et al., 2024; Tong et al., 2023; Lauvaux et al., 2022; Irakulis-Loitxate et al., 2022; Maasakkers et al., 2022; Cusworth et al., 2021), because these sectors offer the highest potential for emission reductions (Nisbet et al., 2025; Rogelj and Lamboll, 2024). In addition, large point sources, e.g. from a leaking pipeline or a landfill, usually produce well-defined, localized plumes with GHG enhancements above instrument detection limits.

In fact, agriculture is the largest single anthropogenic source of both $CH_4$ and $N_2O$ emissions, according to BU and TD estimates. Agriculture accounts for approximately 40 % of total anthropogenic $CH_4$ emissions and 56 % of $N_2O$ emissions (Saunois et al., 2024; Tian et al., 2024). However, our current observational capabilities are still very limited in terms of constraining areal and spatially complex emissions, such as from agriculture (or wetlands). Even if the emission totals are large, they are hard to detect because emissions are dispersed over larger areas producing only small GHG gradients. Agricultural

$CH_4$ emissions vary widely in both space and time, ranging from diffuse areal sources like grazing livestock to localized point sources such as manure heaps or slurry lagoons, with temporal fluctuations driven by feeding patterns, weather, and seasonal farming practices (Nisbet et al., 2025; Laubach et al., 2024; Saunois et al., 2024; Carranza et al., 2022; Kelly et al., 2022; Morgavi et al., 2010; Simpson et al., 1995). $N_2O$ emissions from agriculture, produced through nitrification (Bremner and Blackmer, 1978) and denitrification (Firstone, 1989), are shaped by both spatial patterns — diffuse release from fertilized soils

versus concentrated output from manure or slurry — and temporal dynamics driven by soil moisture changes, temperature shifts, and the timing of fertilization or planting (Kang et al., 2025; Eckl et al., 2021; Butterbach-Bahl et al., 2013; Chadwick, 2005; Sommer et al., 2000).

The high spatio-temporal variability of agricultural emissions presents a challenge for comprehensive top-down quantification. As highlighted by Laubach et al. (2024) and Nisbet et al. (2025), a wide range of measurement techniques are employed

to quantify agricultural emissions, including mass-balance approaches, flux chambers, gradient methods, inverse-dispersion modeling, and tracer-ratio methods. These approaches are applied from various stationary and mobile platforms. However, most of these techniques are limited to small spatial scales and localized settings. There have also been first attempts to detect agricultural $CH_4$ emissions from space, despite the typically weak atmospheric enhancements. Although promising, these satellite systems have yet to be tested (Bukosa et al., 2024). For $N_2O$, there is currently no operational satellite-based mon-

itoring capability; however, ongoing research efforts indicate future potential for space-based detection (Kiemle et al., 2024; Ricaud et al., 2021). On the regional-scale, first airborne studies have demonstrated the feasibility of quantifying agricultural emissions (Dacic et al., 2024; Eckl et al., 2021; Yu et al., 2021; Hiller et al., 2014; Wratt et al., 2001). Those studies either rely on a supporting modeling framework to interpret the observations or measure integrated fluxes over broad areas, often without resolving small-scale emission hotspots.

In summary, methods for quantifying agricultural emissions at regional to continental scales remain scarce or highly dependent on supporting information. The spatial heterogeneity of agricultural sources requires systems capable of capturing



large-scale areal fluxes without sacrificing the resolution needed to detect localized emission hot spots. Eddy covariance (EC) is a powerful tool for quantifying GHG fluxes from areal sources, without relying on auxiliary data (Laubach et al., 2024; Foken, 2021; Morin, 2019; Haszpra et al., 2018). When applied from aircraft, EC also provides spatial coverage and insight
into flux distribution patterns at fine resolution (Shaw et al., 2022; Vaughan et al., 2021; Wilkerson et al., 2019; Wolfe et al., 2018; Kohnert et al., 2017; Metzger et al., 2013; Kiemle et al., 2011, 2007). In this study, we present a $N_2O$-optimized airborne EC setup to quantify agricultural $CH_4$ and $N_2O$ emissions, with high spatial resolution, to our knowledge, a novelty for $N_2O$. This paper is structured as follows: Section 2 outlines the EC measurement principle, data processing steps, and quality control procedures. It also introduces the airborne EC instrumentation and describes the flight strategy during the GHGMon (Green-
house gas monitoring) aircraft campaign. Section 3 evaluates the performance of the EC system. Section 4 presents first flux results for $CH_4$ and $N_2O$ over Friesland, a region in the Netherlands characterized by intensive agricultural activity , providing insights into emission processes (Van Der Heide et al., 2011). We report regional-scale fluxes of $N_2O$ and $CH_4$ observed during the growing season, including a dry-to-wet transition, and demonstrate the system's ability to detect both emission hotspots and hot-moments. We further compare the results with other agricultural regions and two BU emission inventories.

## 100  2   Methods

We conducted airborne EC flux measurements with the DLR research aircraft Cessna Grand Caravan 208-B. The aircraft was equipped with a 10 Hz GHG analyzer (MIRO Analytical AG, Wallisellen, Switzerland), which was specifically adapted for the airborne deployment, as well as with the aircraft's standard equipment for meteorological measurements including vertical wind, called METPOD (Mallaun et al., 2015). This section introduces the principles of Eddy Covariance measurements and the
challenges related to the airborne application of this method. It presents the specifications of the GHG analyzer, it's airborne version and the meteorological measurements, as well as the flight strategy of the GHGMon campaign.

### 2.1   Eddy Covariance Analysis

EC measures the covariance between the vertical wind $w$ and a scalar quantity $c$, directly quantifying the vertical fluxes of surface emissions into the planetary boundary layer (PBL) under the assumption of well-developed turbulence, ergodicity, and
spatial homogeneity of the source area (Foken, 2021; Kaimal and Finnigan, 2020; Stull, 1988). While homogeneity implicitly restricts EC to areal sources such as pastures or croplands, airborne EC can also capture emissions from clusters of point sources like individual farms by flying at higher altitudes and greater distances, effectively merging their emissions into a single, homogeneous source through spatial averaging (Yuan et al., 2015). The assumption of ergodicity implies equivalence between spatial and temporal means, which is not always fulfilled in real-world conditions. Violations of ergodicity often arise
from changing weather conditions, such as frontal passages, or from submeso-scale motions like gravity waves (Stefanello et al., 2020). Aircraft-based EC is particularly sensitive to these submeso-scale distortions because of the large spatial extent of flight legs.



### 2.1.1 Vertical turbulent fluxes

Turbulent motion can lead to a non-zero transport of quantities such as GHGs or particles, even if the mean flow, meaning the mean wind speed, is zero. If the mean vertical wind speed, $\overline{w} = 0$, and horizontal advection as well as storage are negligible, the fluctuations in the vertical part of the flow dominates and this turbulent eddy transport can be measured to quantify exchange processes between the earth's surface and the PBL (Stull, 1988). The flux of a scalar quantity $c$ (for example $T$, $H_2O$ or GHGs) is then computed as:

$$F = \rho_{\text{air}} \overline{w'c'}, \tag{1}$$

where $\rho_{\text{air}}$ is the air density, $w'$ and $c'$ are the instantaneous deviations from the mean values $\overline{w}$ and $\overline{c}$, respectively. The fluctuating parts $x'$ can be isolated from the means via Reynolds decomposition of the time series of a signal $x$:

$$x = \overline{x} + x' \tag{2}$$

Furthermore, $\overline{w'c'}$ denotes the covariance between $w$ and $c$:

$$\overline{w'c'} = \frac{1}{N} \sum_{i=0}^{N-1} (w' - \overline{w}) \cdot (c' - \overline{c}) \tag{3}$$

where $N$ is the number of samples taken in the measurement segment (flight path), for which the flux is calculated (Foken, 2021). These measurement segments, in the further course called legs, must be chosen in order to comply with the EC assumptions and balanced between high spatial resolution and low uncertainties. Figure 1 illustrates the eddy covariance flux principle between $N_2O$ and $w$ for a example leg of about 7.5 km in length. Updrafts (positive $w'$) coincide with increased $N_2O$ concentrations (positive $c'$), while downdrafts align with lower concentrations, resulting in a net positive vertical flux.

Flux quantification with eddy covariance relies on three key assumptions: stationarity, horizontal homogeneity, and well-developed turbulence. These assumptions ensure that only the vertical turbulent transport term in the governing equations contributes significantly to the measured flux. Stationarity implies that there is no net accumulation or loss of scalar quantities within the planetary boundary layer (PBL), effectively excluding storage effects from the total flux. Horizontal homogeneity assumes that the landscape and flow field do not vary in the horizontal direction, which eliminates the influence of advection and horizontal turbulent transport.

When using EC fluxes for surface emission mapping or inventory comparisons, in situ fluxes, measured at a certain altitude $z_{\text{meas}}$, must be scaled to represent surface fluxes. GHG fluxes are primarily driven by surface processes, but this influence diminishes with altitude. At the top of the PBL, fluxes are largely controlled by exchange processes between the PBL and the free troposphere in the entrainment zone. As a result, fluxes can approach zero when concentrations gradients between PBL and free atmosphere are minimal or, may even change sign at or just below $z_i$, the boundary layer height. For example, the sensible heat flux, $H$, can become negative in the presence of temperature inversions, and GHG-fluxes may reverse if the concentration in the free atmosphere is elevated through long-range transport (Gioli et al., 2004; Vickers and Mahrt, 1997; Stull, 1988)). We accounted for this vertical flux divergence by measuring fluxes at different altitudes inside the PBL, and linearly extrapolating



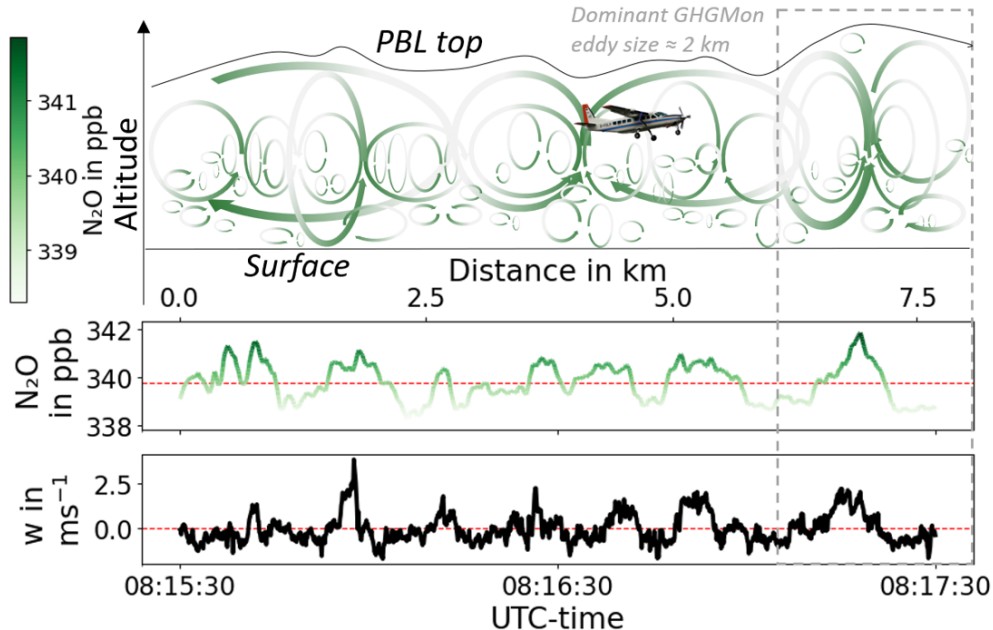

**Figure 1.** Schematics of EC measurements. The sketch in the top visualizes eddies of different sizes, building the turbulent boundary layer. Enhanced concentrations from surface emissions (greenish) are transported upwards in updrafts, lower concentrations (grayish-white) coincide with sinking air parcels. The aircraft-based EC system must cover all relevant contributions from different sized eddies. The lower subplots show $N_2O$ and $w$ time series ($\approx 2$ min) from the forenoon flight on 21 June 2023. Means are marked as red dashed lines. The Pearson coefficient $R^2$ for this leg is 0.46. The eddies contributing most to the flux had typical scale-sizes of $\approx 2$ km for the GHGMon flights.

them to the surface. Corresponding calculations and related uncertainties can be found in Appendix A. Footprint calculations
are necessary to accurately assign localized flux measurements (at the position of the aircraft) to corresponding source areas on
the surface. Kljun et al. (2004) provides a 1-D parameterization technique to estimate the footprint extent considering the along
wind dispersion, whereas Metzger et al. (2012) expanded this model by a gaussian modeled cross-wind contribution function,
to get an estimate of the 2-D footprint area. Both models were applied in different studies, including airborne EC (Vaughan
et al., 2021, 2017; Metzger et al., 2012). Additionally, there are simple approximations of the half-width footprint size, used
by Wolfe et al. (2018) or Karl et al. (2013), which is defined to contain 50 % of the measured flux:

$$d_{0.5} = 0.9 \frac{u_{\mathrm{mean}} \cdot z_{meas}^{2/3} \cdot z_i^{1/3}}{w*} \tag{4}$$

Here, $u_{\mathrm{mean}}$ is the mean PBL wind speed and $w*$ is the convective velocity (with gravitation acceleration $g$, and virtual
temperature $\theta_v$):

$$w_* = \left( \frac{g}{\theta_v} \overline{w'\theta_v'} z_i \right)^{1/3} \tag{5}$$



Since a more in-depth inventory comparison with our fluxes, as well as an explicit identification of emission hotspots on the surface is part of an upcoming study, we used Equation 4 to estimate footprint sizes for the flight planning (see Section 2.3), but did not calculate more complex footprints.

### 2.1.2    Data quality and processing

Before computing and reporting EC fluxes, as outlined in Section 2.1.1, we prepared the dataset, performed quality control,
and assessed its suitability for applying the EC theory and the underlying assumptions.

Preprocessing for EC flux calculations involved multiple steps. We began by inspecting the data for gaps and missing rows to identify potential recording malfunctions. Only one such instance was detected: a single $10 \, \mathrm{min}$ gap during one flight. The GHG analyzer clock was synchronized with METPOD's GPS time every $60 \, \mathrm{s}$ using the IZ2BKT software. Any remaining single empty GHG measurement data point— only a few per flight — were linearly interpolated. Calibration intervals, which were
necessary every $10-15 \, \mathrm{min}$ (see Section 2.2.1) were excluded from the interpolation and treated as natural limitations for the measurement segment (leg) length, for which the flux is to be calculated. The second step included the choice of these segments. Due to the calibration intervals, single flux legs had a maximum duration of around $10-15 \, \mathrm{min}$, corresponding to a mean leg length ($L$) of $\approx 37-56 \, \mathrm{km}$. Flux calculations for long legs result in smaller turbulence-related errors (see Appendix A), but they are also more likely to violate the fundamental assumptions of EC, e.g. because of advection or surface inhomogeneities.
We have chosen our leg lengths with respect to typical time and length scales used in ground-based EC measurements. Typical averaging periods for surface or tower-based EC fluxes are $30 \, \mathrm{min}$ (Karimindla et al., 2024; Murphy et al., 2022; Velasco et al., 2005), corresponding to length scales of $5-11 \, \mathrm{km}$ for average horizontal wind speeds of $3-6 \, \mathrm{m \, s^{-1}}$, which translates to $\approx 90-180 \, \mathrm{s}$ flight time with the campaign average aircraft speed of $62.1 \, \mathrm{m \, s^{-1}}$. Individual leg lengths varied based on calibration intervals, aircraft maneuvers, and other factors. The leg-averaged fluxes were calculated as moving windows of at
least $90 \, \mathrm{s}$ in length with a step size of $10 \, \mathrm{s}$ (corresponding to $\approx 620 \, \mathrm{m}$ spatial displacement) to have a large overlap between the legs and guarantee high spatial resolution. For each leg, the lag-time between the GHG analyzer and the METPOD system was inferred using the maximum cross-corelation method between both instruments, yielding reproducible $2.0-2.1 \, \mathrm{s}$. Then we removed linear trends from each leg. Despiking was not applied, as the recorded data were already quality-controlled. Likewise, no water vapor density correction (Foken, 2021) was necessary, since mole fractions were reported relative to dry
air (see Section 2.2.1). Legs with changes in altitude of $>50 \, \mathrm{m}$ as well as legs above $z_i$ were excluded. Furthermore, all legs with maximum roll angles exceeding $5°$ were excluded, as aircraft roll cause errors in vertical wind speed measurements with the five-hole probe (Mallaun et al., 2015). Legs with negative sensible heat fluxes were also excluded from GHG flux computations, as they typically indicate down-mixing of an inversion (e.g. during nighttime) or mixing between the PBL and free troposphere— both scenarios are incompatible with measuring instantaneous GHG surface fluxes. These situations were
rare but occurred in some of the first legs of flights with early start times. Furthermore, the convective velocity was used to flag legs with weak turbulence conditions: $w^* < 0.5 \, \mathrm{m \, s^{-1}}$. Single leg limits of detection (LODs) were computed (see Appendix A), but were not always used to flag fluxes below the corresponding LODs, since those fluxes can still provide meaningful information when averaged over multiple overpasses (Langford et al., 2015).



### 2.1.3 Low- and high-pass filtering and correction

Spectral and cospectral analysis are essential tools in EC studies, as they provide insights into the scales of turbulent transport and the potential loss of fluxes due to measurement limitations. In airborne EC, the fast-moving platform introduces additional challenges, such as sensor response times, spatial averaging, and possible platform motion effects, which can dampen high-frequency fluctuations. Mallaun et al. (2015) showed, that no significant limitations in wind, temperature and humidity data with a 10 Hz resolution arise from the meteorological instrumentation. For the GHG analyzer, we can compare spectra and

cospectra with those of the METPOD system, allowing us to identify potential high-frequency flux losses or white-noise contributions due to limitations of the GHG analyzer and verify optimal time synchronization between the two instruments. Furthermore, spectral analysis helps to identify the inertial (turbulent) subrange, where power spectral densities of scalars and wind are in agreement with the -5/3 law proposed by Kolmogorov (1991). The cospectral analysis focuses on the covariance of the vertical wind velocity with scalar quantities (e.g., temperature or trace gas concentration) to verify whether the fluxes

are accurately measured across all relevant scales. This step is critical for assessing whether the observed fluxes include purely turbulent contributions or also low-frequency variability, associated with larger atmospheric structures, which possibly cause transport, which is not representative of surface fluxes anymore. Studies such as Metzger et al. (2013) and Mann and Lenschow (1994) highlight the importance of correcting for spectral attenuation to avoid underestimating fluxes, especially in cases where sensor limitations or data processing truncate certain frequencies. We applied spectral and cospectral analysis to achieve robust

flux calculations and improve the reliability of surface-atmosphere exchange estimates. Both analyses were conducted on the GHGMon data using the scipy.fft module in Python.

If EC fluxes should be representative of the instantaneous ecosystem exchange rate of scalars, the complete capture of all relevant turbulent scales into the flux calculation is essential. However, this integration can be compromised in several ways. First, the limited instrument sample frequency can act like a low-pass filter and lead to significant loss of flux contributions in

the high-frequency domain (Herig Coimbra et al., 2024; Ibrom et al., 2007). These small-scale fluctuations can be dampened by long inlet tubing combined with a small flow rate, or simply not detected, when the data acquisition rate is too low or the analyzer's precision is not sufficient. In closed-path systems, they can be smeared out in the cavity due to limited sample turnover time (Metzger et al., 2016). With 15.3 sLm (standard liter per minute) flow rate and 1/2" tubing at a measurement frequency of 7 Hz, the mixing in the GHG analyzer's cavity and the limited precision of the analyzer cause the main high-

frequency loss of our system, while tube effects are small. Horst (1997) proposed a simple equation to estimate high-frequency attenuation of response-time-limited sensors:

$$\frac{F_{\mathrm{meas}}}{F_{\mathrm{true}}} = \frac{1}{1 + 2\pi f_{\mathrm{m}}\tau} \tag{6}$$

Here, the ratio of measured (low-pass filtered) flux $F_{\mathrm{meas}}$ to true flux $F_{\mathrm{true}}$ depends only on the frequency of maximum covariance, $f_{\mathrm{m}}$ and the response time of the analyzer, $\tau$. In laboratory experiments, we determined the response time to be

0.16 s, independent of the measured parameter. $f_{\mathrm{m}}$ can be inferred via measured cospectra of fast METPOD $H_2O$ and $w$. Equation 6 can be used to correct measured fluxes according to their high-frequency loss, or to define this loss as part of the



flux error ($\sigma_{\mathrm{HF}} = \frac{F_{\mathrm{meas}}}{F_{\mathrm{true}}}$), which should be taken into account in the assessment of flux uncertainty. We chose the second option, as discussed in Appendix A.

In addition to the unwanted low-pass filtering in the system, high-pass filtering by inappropriate (too short) leg lengths
can lead to systematic underestimation of fluxes due to missing large-scale contributions. This effect is also discussed in Appendix A and included in the uncertainty analysis (Equation A1). Furthermore, and especially for aircraft EC measurements, unwanted, non-turbulent large-scale contributions have to be considered. Those are more likely to appear in airborne than ground-based EC, because of the larger footprint areas. These mesoscale influences can increase or decrease the calculated flux, making it essential to carefully determine the optimum leg length (Foken, 2021). Detrending or block averaging can
address this issue, but they require additional constraints, since selecting an appropriate window size for block averaging is not straightforward. Detrending can be performed in a simple first-order (linear) approach, or by removing more contributions with increasingly higher polynomial degrees of the filter. Filtering can also be applied in the Fourier space, but the problem remains to define an appropriate cut-off frequency, separating between turbulent and non-turbulent, larger scales. Another popular approach to determine the optimum leg length is the ogive method (Sun et al., 2018), which infers the lowest frequency that
needs to be included in EC analysis from the cumulative frequency contributions. When ogives show no further contributions at large scales (indicating converging flux), this point is defined as lowest frequency, which is necessary for flux calculation. However, this method is primarily useful for identifying potentially missing large-scale turbulent contributions. It is not suitable for cases where mesoscale contributions are present and need to be removed, as the estimation of optimal leg length is based on flux contributions rather than turbulence characteristics. An alternative option is the continuous wavelet transform (CWT),
which resolves fluxes in both the time and frequency domains. The analysis of large-scale contributions and their spatial coherence can separate the largest turbulent scales from the smallest mesoscale contributors (Metzger et al., 2013). However, this coherence criterion also depends on the constraint of a minimum coherence threshold for the turbulent regime. Other studies have discussed the effects of large-scale contributions by comparing fluxes calculated with different leg lengths (Sun et al., 2018; Desjardins' et al., 1988). However, no a-priori cut-off frequency was established to separate turbulent scales (in
the inertial subrange) from the mesoscale, as this threshold depends on atmospheric stability conditions, measurement height, and the boundary layer height (Gioli et al., 2004; Kaimal et al., 1972).

The parameter that best reflects the influences of stability and measurement height for EC calculations in unstable conditions is the vertical wind $w$. As stability decreases and measurement height increases, the frequency of maximum energy contribution in the power spectrum shifts towards lower values, and vice versa (Kaimal and Finnigan, 2020). In the inertial subrange, where
turbulence is considered isotropic and the core assumptions of EC for quantifying instantaneous ecosystem flux are valid, the ratio of power spectra for $w$ and zonal wind $u$ should ideally match a value of 4/3, as proposed by Kaimal et al. (1972) from theoretical considerations. In practice, however, observations often deviate from this theoretical value, with many studies reporting values closer to one (Biltoft, 2001). Although this isotropy ratio remains poorly constrained by empirical data, deviations at low frequencies, which originate from non-turbulent motions, can be used to identify the transition frequency
between turbulent and mesoscale processes (Kaimal and Finnigan, 2020). This transition frequency can be deduced from the $w$ spectrum. The frequency of peak power in the $w$ spectrum, $f_{\mathrm{peak}}$, approximately indicates the scale of the largest turbulent





eddies. Frequencies below this threshold typically correspond to sub-meso motions like horizontal meandering, gravity waves, significant advection, or micro fronts. Ideally, a clear spectral gap would appear at frequencies below $f_\mathrm{peak}$, where power drops sharply, enabling a unambiguous separation between turbulent and mesoscale contributions. However, such a gap is often absent in measured spectra, particularly in spectra of variables other than $w$ (Stefanello et al., 2020; Biltoft, 2001).

We used $f_\mathrm{peak}/2$ to define a threshold for turbulent contributions and removed lower frequencies with a $4^\mathrm{th}$-order Butterworth filter (Butterworth and Else, 2018). The division by two provides some tolerance in determining $f_\mathrm{peak}$ and ensures that few, if any, large but genuine turbulent scales are excluded from flux calculations. For the GHGMon flights, this $f_\mathrm{peak}/2$ threshold showed excellent agreement with the onset of isotropy (4/3 or unity ratio of $w$ and $u$ spectra) towards higher frequencies. An example is shown in Figure 2, where the power spectra of $w$ and $u$ as well as their ratio for the flight on 14 June 2023 are displayed. There is a clear, sudden increase in the spectral ratio (right sub-panel), starting from near zero at low frequencies—where $u$ fluctuations dominate over $w$, indicating the absence of PBL-related vertical turbulence—rising to values above 4/3 at the lowest turbulent frequencies, and then gradually decreasing to around one at the high-frequency end. Similar observations have been documented for unstable PBL stratification in Biltoft (2001).

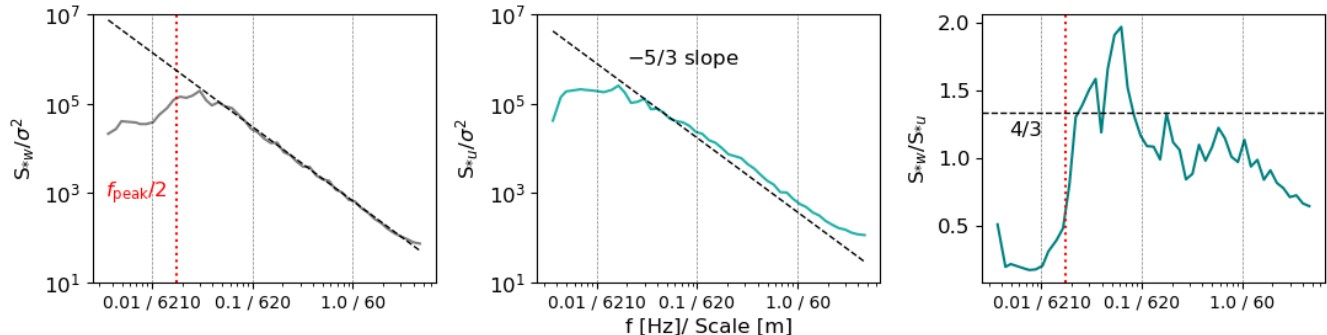

**Figure 2.** Averaged power spectra, $S_*$ normalized by variance ($\sigma^2$) for $w$ and $u$ wind components on 14 June 2023. The black dashed lines in the first two subplots indicate the -5/3 slope, expected in the inertial subrange. The red dotted vertical line in the first plot marks the $f_\mathrm{peak}/2$ frequency. The third panel shows the ratio $S_{*w}/S_{*u}$, which approaches the theoretical isotropy value of 4/3 for turbulence in the inertial subrange for frequencies larger than $f_\mathrm{peak}/2$. For clearer presentation, spectra were bin-averaged with 0.2 Hz. Note the ticks on the abscissa are either frequency or scale size, separated by a slash symbol.

## 2.2 The airborne eddy covariance measurement system

### 2.2.1 GHG analyzer

Successful airborne EC deployments require not only fair-weather, turbulent conditions but also highly sensitive instrumentation capable of detecting subtle GHG fluctuations far from the source. Quantum cascade laser spectrometry (QCL) has proven to be well suited for high-precision airborne $CH_4$ and $N_2O$ measurements (Kostinek et al., 2019; Santoni et al., 2014). We used





a commercially available MIRO MGA[3] QCL-based absorption spectrometer to measure $N_2O$ and $CH_4$. The instrument also records mole fractions of carbon monoxide (CO) and water vapor ($H_2O$). It operates with two mid-infrared lasers covering the wavenumber ranges $2190.0-2190.4$ cm$^{-1}$ and $1281.4-1281.7$ cm$^{-1}$. The lasers are centered in their wavenumber domains with individual thermoelectric coolers and modulated by a time-multiplexed intermittent continuous-wave driving scheme. The single pulse durations are between $10-100$ µs and the overall modulating period takes about $0.5-1$ ms. These current pulses

cause a heating of the lasers, resulting in frequency modulation. The detected spectra are fitted based on spectral line transmission data inferred from the HITRAN database and converted to molefractions using Lambert-Beers law (MIRO Analytical AG, 2021; Gordon et al., 2022). The sample gas streams through the 0.5 L cavity driven by an external vacuum scroll pump (oil-free Anest Iwata ISP500C, with an aircraft-adapted Baumüller electric motor) at $\approx 15$ sLm. The optical cell is regulated at a pressure of 73 hPa with an Alicat Scientific pressure controller. A thermoelectric recirculating chiller (Solid State Cooling

Systems, model Thermorack 401) keeps the lasers and the optics compartment at their operating temperatures. Figure 3b shows the rack of the analyzer installed in the aircraft.

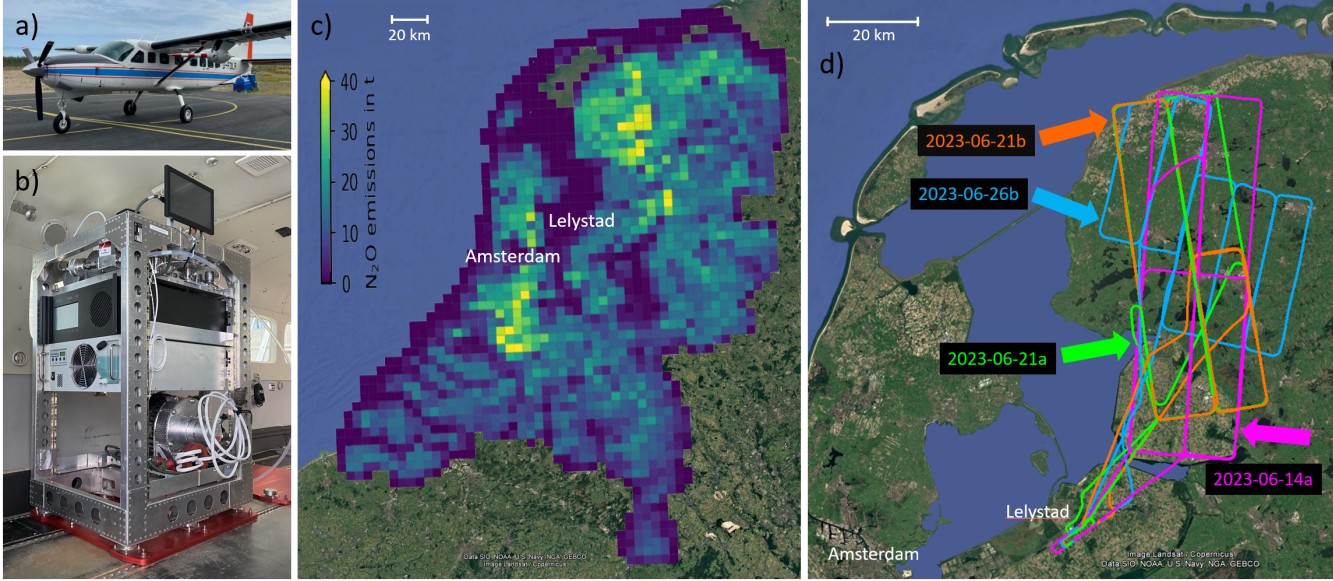

**Figure 3.** a) DLR research aircraft Cessna Grand Caravan 208-B with the METPOD mounted under the left wing. b) GHG analyzer setup, with MIRO MGA[3], thermoelectric chiller, scroll pump, calibration unit, and periphery for in-flight access and control. c) Agricultural $N_2O$ emissions of the Dutch national inventory Emissieregistratie per km[2] per year. Agricultural emissions account for the vast majority of total $N_2O$ emissions ($>> 90$ %) in the northern Netherlands, especially for the region shown in panel d): Flight tracks of four selected flights over Friesland. Arrows indicate corresponding mean PBL wind directions. Panels c) and d) use Google Earth imagery (Image © Landsat/Copernicus, Maxar Technologies).

Before the campaign we performed laboratory tests with reference gas to investigate the performance of the analyzer. 10-Hz data of reference gas measurement exhibit a $1\sigma$ precision of 4.3 ppb for $CH_4$ and 0.15 ppb for $N_2O$, which further improves to



1.5 ppb and 0.05 ppb, respectively, for resampled 1-Hz data. The manufacturer reports similar specifications for the analyzer
(MIRO Analytical AG, 2021). The instrument provides particularly high precision for $N_2O$, outperforming other airborne
systems (Kostinek et al., 2019; Wilkerson et al., 2019; Santoni et al., 2014), making it well-suited for EC measurements.
During several laboratory tests, we observed a linear drift in most cases, which was of different direction but never exceeded
$5.3\,\mathrm{ppb\,h^{-1}}$ for $CH_4$ and $0.2\,\mathrm{ppb\,h^{-1}}$ for $N_2O$. Due to the linear nature of the trend, it can be effectively removed through
linear detrending, minimizing any impact on EC measurements. However, since the data are intended for broader application
beyond EC, an in-flight calibration procedure was implemented, with calibration intervals every $10-15\,\mathrm{min}$, to ensure the
highest possible accuracy, when possible, in line with recommendations from the World Meteorological Organization (WMO)
(Crotwell, 2018).

**In-flight calibration**

The calibration unit consists of two 2 L reference gas cylinders, which are connected with 1/2" and 1/4" PTFE (polytetrafluo-
roethylene) and stainless steel tubing with the reference gas port of the analyzer. The pressure reducers decrease the pressure
pf 120 bar of the gas cylinders to 3 bar and are permanently open during a flight and each is equipped with a stainless steel
solenoid valve (SMC JSX) for opening or closing. A Bronkhorst mass flow controller (MFC) regulates the calibration gas flow
to 18 sLm, ensuring an overflow of at least 3 sLm into the cabin. This prevents contamination and guarantees clean, reliable
calibrations. The 2 L reference gas cylinders are measured for 20 s each at irregular intervals, approximately every $10-15\,\mathrm{min}$
during the flight (manually started by the onboard operator, when possible during aircraft turns, descents and climbs to avoid
losing EC measurement time). They are refilled from two larger 50 L reference cylinders before every flight. One of the gases
serves as low standard , the other as high standard for a two-point calibration. The large cylinders were cross-calibrated against
NOAA WMO standards before and after the campaign. Due to the small concentration difference between the low and high
$N_2O$ standards ($< 2$ ppb) —resulting from limited manufacturing accuracy—a two-point calibration using linear regression
was not feasible for $N_2O$. Instead, we applied an offset correction. Linear interpolation was used between consecutive calibra-
tions.

**GHG water vapor correction**

The standard output files of the MIRO analyzer contain mole fractions of $N_2O$, $CH_4$ and CO. Generally, GHG mole fractions
are reported in relation to the dry gas molecules to enable unrestricted comparability. Hereby, the dilution effect and quantum
mechanical effects related to spectral line broadening have to be considered. Both can alter GHG recordings and are not
accounted for in the spectral fitting software of our MIRO MGA[3] instrument. Extensive laboratory tests revealed a dependency
of the measured GHG mole fractions on $H_2O$ levels and were used to elaborate an accurate correction of these effects. Details
on the experimental setup, the derivation of the correction curve, and its validation will be presented in a separate publication.
In Appendix B, the applied water vapor correction is validated by comparing the corrected $N_2O$ and $CH_4$ measurements with
data from the independent reference instrument JAS (see Section 2.2.3). Furthermore, possible inaccuracies of the water vapor
correction are accounted for in the EC uncertainty assessment (see Section 3.2).



### 2.2.2 Meteorological measurements

The DLR Cessna is equipped with the METPOD (meteorological sensor package) and blackMAMBA (measurement acquisition of meteorological basics) systems for meteorological observations. Part of the former is the 2 m long nose boom, with a Rosemount 858AJ five-hole probe for high-precision measurements of the true airspeed which is crucial for the 3-D wind calculation. Furthermore, the METPOD, mounted under the left wing of the aircraft, contains pressure, temperature, and humidity sensors. The blackMAMBA system includes the data acquisition unit and a time server (Mallaun et al., 2015), which assures time synchronization between the meteorological and GHG data recording via the BktTimeSync software (IZ2BKT). The vertical wind component is measured with a time resolution of 100 Hz and an uncertainty below $0.2 \, \mathrm{m \, s^{-1}}$. Humidity is measured by a Ly-$\alpha$ absorption hygrometer (Model L5, Buck research) with an uncertainty of 2%, while temperature is recorded by an open-wire PT100 with an uncertainty of 0.15 K. Detailed information on the meteorological measurements and their performance is provided in Mallaun et al. (2015).

### 2.2.3 Aircraft setup and additional payload

For the GHGMon campaign, we installed the GHG analyzer, thermo-chiller, and pump in a 19" standard aluminum rack designed for aircraft integration. We placed the calibration unit on top of the rack to allow fast and easy exchange of calibration cylinders. To minimize mechanical stress and prevent misalignment of optical components from vibrations, we mounted the entire rack on shock absorbers. A customized power unit supplied all electronic devices and converted the board voltage (28 VDC) to the required output (12 V for MFC and monitor, 24 VDC for pump control, fan and valves). The power supply of the GHG analyzer was changed from 230 V to enable direct use of the on-board voltage. The trace gas inlet for the analyzer is rearward-facing and located in a wing pod under the right wing of the aircraft. We used a 6 m, 1/2" PTFE tube to direct the sample gas to the instrument in the cabin. Before reaching the analyzer, the gas passed through a 1 μm PTFE filter to remove particles and maintain the purity of the measurement cell. The filter restricted the flow to approximately 13 sLm, reducing the effective measurement frequency to 7 Hz. The METPOD system is part of the standard configuration of the DLR Cessna and its integration is explained in Mallaun et al. (2015). The wind and meteorology sensors, positioned under the left wing, are approximately 8 m away from the GHG inlets. This displacement limits the EC system's ability to resolve the smallest scales, which turned out to not contribute significantly to the flux, as shown in Section 3. Another instrument aboard the Cessna was the Jena Air Sampler (JAS). This system collects discrete air samples in 12 1-L glass flasks during the flight. Post-campaign laboratory analysis at the Max Planck Institute for Biogeochemistry in Jena provided information on mole fractions of $CO_2$, $CH_4$, $N_2O$, $H_2$, $SF_6$, CO, $O_2/N_2$ and $Ar/N_2$ ratios, as well as $\delta^{13}C(CH_4)$ and $\delta^2H(CH_4)$. A detailed description of the instrument is given in Gałkowski et al. (2021). One major benefit of JAS is the possibility of evaluating data quality for new instrument setups. JAS has already been deployed in several studies (Gałkowski et al., 2021; Fiehn et al., 2023) and provides a low uncertainty of 0.13 ppb for $N_2O$, establishing it as a reliable standard for comparison with the MIRO GHG analyzer.





## 2.3 GHGMon campaign & flight strategy

The GHGMon campaign took place from 10 to 30 June 2023, the campaign base was at the Lelystad airport (EHLE),
Flevoland, the Netherlands. A total of 14 research flights, usually lasting $2.5-3$ h, were carried out, each covering a distance
of $\approx 500-600$ km. The average flight speed was $\approx 62.1$ m s$^{-1}$ and individual legs were flown between FL08 (243 m amsl.,
due to national flight altitude restrictions) and the top of the planetary boundary layer (PBL). During each flight, at least two
vertical profiles were flown to determine the boundary layer height and measure gradients into the free troposphere up to
$\approx 3000$ m amsl.. Since the primary focus of the campaign was GHG flux measurement using EC, most flight legs were straight
and maintained at a constant altitude. This is important to assure relatively stable flux footprints and to not violate ergodicity
with changes in altitude. We flew grid-like patterns with most legs perpendicular to the mean PBL wind direction and some
spacing in between, with the aim to cover the entire target area and having high footprint coverage, while still having some
footprint overlap between two parallel legs, to assess consistency of fluxes of different legs. Typical values of the half-width
footprint size $d_{0.5}$ (see Equation 4) ranged between $1-10$ km for the GHGMon flights, therefore we set the spacing between
two parallel legs to $10-20$ km. Some of the legs were repeated to assess temporal flux consistency (see 4.1) and to study
areas of high interest, especially during different meteorological situations. Some legs were varied in altitude to analyze the
vertical flux divergence. For this purpose, we chose legs over the same area and flew one immediately after the other in order
to avoid spatial or temporal flux variations affecting our vertical flux divergence measurements. Well-developed turbulence is
a prerequisite for EC, hence flight days were restricted to fair weather conditions with little to no cumulus cloud cover, using
meteorological forecast data of $z_i$, cloud cover, wind direction and speed, as well as precipitation (and further parameters)
from the European Centre for Medium-Range Weather Forecasts, ECMWF. Flight tracks were planned to cover areas of high
agricultural activity, based on the Dutch national greenhouse gas emissions inventory Emissieregistratie (Emissieregistratie,
2025). It provides bottom-up emission estimates of various species with high spatial resolution (5x5 km for GHGs) and is the
base for national and international emission reporting. Emissions are estimated based on reportings from industry, statistical
data, collected by national institutes and universities (e.g. livestock number or amount of used fertilizer), and in the case of
agriculture with the help of the National Emission Model for Agriculture, which uses IPCC conform Tier $1-3$ approaches with
partially country specific emission factors. Figure 3c shows agricultural $N_2O$ emissions from Emissieregistratie, which account
for more than 90 % of $N_2O$ emissions in Friesland. Figure 3d displays the tracks of four research flights carried out mainly in
the province of Friesland, covering the same area under different meteorological conditions. Those results will be discussed in
more detail in the following sections to analyse the temporal variability of $N_2O$ and $CH_4$ emissions. Table 1 provides additional
meteorological and flight-specific information on those four flights. All of them were conducted with the altitude of all EC legs
well beyond $z_i$, during warm days with moderate to fresh and stable wind conditions. Notable differences in precipitation
occurred during the 48 h before the flights, with dry conditions on 14 and 26 June, and wet conditions during the two flights
on 21 June.





**Table 1.** Research flight overview and meteorological information of the

| Date | time (UTC) | FL in m amsl. | $z_i$ in m amsl. | $T_{\mathrm{mean}_{\mathrm{surf}}}$ in K | $u_{\mathrm{mean}}$ in ms$^{-1}$ / WA in ° | 48 h prec. in mm |
|---|---|---|---|---|---|---|
| 14 June | 07:50−10:35 | 350−396 | 600 | 295.9 | 7.9 / 92 | 0.0 |
| 21 June | 06:50−9:45 | 283−520 | 560 | 296.2 | 5.7 / 240 | 11.8 |
| 21 June | 12:00−15:00 | 298−400 | 890 | 297.1 | 7.1 / 254 | 11.9 |
| 26 June | 12:30−15:15 | 263−371 | 1050 | 293.6 | 10.2 / 281 | 0.2 |

## 3 Evaluation of the performance of the airborne EC setup

The following section evaluates the overall performance of the GHG analyzer, flux uncertainties, and spectral characteristics of individual GHGMon flights, to highlight the systems' strengths and limitations.

### 3.1 Spectral analysis

Cospectral information gives insight into the distribution of relevant flux-contributing scale sizes, whether the resolution is insufficient to resolve small turbulent eddies or flux distortion due to large-scale non-turbulent motions. Flight-averaged, normalized cospectra of the scalars with $w$ and normalized, integrated cospectra (ogives) for the flight on 14 June 2023 are shown in Figure 4. Therein, the top row presents results for long leg lengths (15400 m), the bottom row for short leg lengths (8400 m). In the left column, data are not filtered, in the right column, data of both leg lengths are filtered according to the high-pass filtering method based on $S_{*w}$, as discussed in Section 2.1.3. $f_{\mathrm{peak}}/2$ as well as ogive thresholds of 5 % and 95 % are marked. $f_{\mathrm{peak}}/2$ (in this case 0.017 Hz) indicates the frequency of maximum power in the spectrum of $w$ and gives an estimate on the transition between turbulent and mesoscale motions. We used this as cut-off for the high-pass filtering. The 5 % and 95 % percentiles mark the frequencies, for which contributions to the ogives (and hence to the flux) are below 5 % and above 95 %. All four cases exhibit positive cospectra, indicating upward flux, with dominant contributions occurring between $\approx 0.015$ Hz and $\approx 0.3$ Hz for all scalars. This corresponds to characteristic eddy sizes between 4100 m and 200 m, for an average true air speed of the aircraft of 62.1 m s$^{-1}$. The 5 % and 95 % cumulative flux thresholds show only minor differences between the unfiltered long and short legs (left column), indicating that the low-frequency end of the turbulent spectrum was fully captured. If this were not the case, longer legs would have revealed stronger contributions at lower frequencies. Hence, our choice of the short leg length $L$ is sufficient to resolve the largest flux-contributing eddies. Any potential minor under-representation of these largest eddies is accounted for in the systematic uncertainty term $\sigma_{\mathrm{syst.turb.}}$ (see Appendix A). For the high-pass filtered cases, the 5 % and 95 % flux contribution thresholds are slightly shifted towards higher frequencies. In the unfiltered data, less than 5 % of the ogives originates from frequencies below 0.014 Hz, corresponding to eddy scales of approximately 4400 m. In contrast, for the high-pass filtered data, less than 5 % of the flux stems from frequencies below 0.016−0.017 Hz (scale size $\approx 3650$ m). This marginal shift of dominating eddy scale sizes after applying the high-pass filter indicates that contributions from non-turbulent, large-scale motions are negligible. Since the frequency ranges of the main flux contributions in





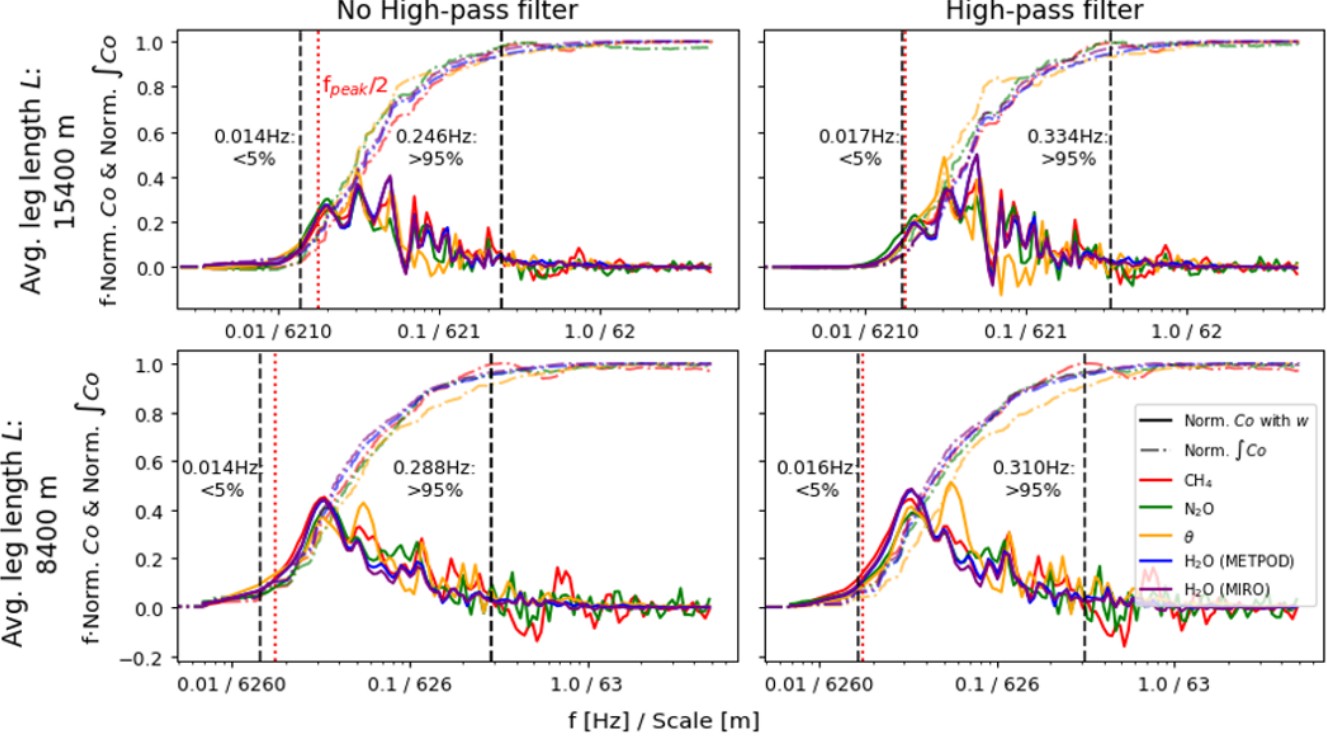

**Figure 4.** Cospectra for the flight on 14 June 2023. The top row shows frequency-multiplied, normalized flight-averaged cospectra of $N_2O$, $CH_4$, $\theta$, and $H_2O$ of the GHG analyzer and METPOD with the vertical wind $w$ in 25-fold exaggeration for $L = 15400$ m. Dash-dotted lines are corresponding to normalized cumulative cospectra (ogives) and the vertical black dashed lines mark the average frequency across all species, at which 5 % and 95 % of the flux is reached. The bottom row shows the same for short legs $L = 8400$ m. The left column shows unfiltered data, the right column data which are high-pass filtered with a cut-off at $f_{\text{peak}}/2$. $f_{\text{peak}}/2$ marked as vertical red dotted line. Note the ticks on the abscissa are either frequency or scale size, separated by a slash symbol.

the unfiltered cases remain nearly identical regardless of leg length $L$, the slight shift seen in the filtered data likely results from the limited precision of the high-pass filter rather than from actual sub-mesoscale influences. The purpose of applying the high-pass filter was to serve as an instrument to identify potential distortions in the flux due to sub-mesoscale motions, not the determination of a precise threshold between turbulent eddies and larger-scale motions. No evidence of sub-mesoscale contributions was found in any of the four flights over Friesland.

Beyond insights into low-frequency contributions, Figure 4 also reveals whether high-frequency losses occur due to limitations in the sampling system. Cospectra derived from fast (100 Hz) METPOD data ($H_2O$ and $\theta$) approach zero above $\approx 0.3$ Hz, and normalized ogives rapidly converge to one, indicating that high-frequency contributions to $H$ and $\lambda E$ (latent heat flux) are fully captured. The slower GHG analyzer data shows the same behavior (vanishing cospectra and ogives close to one) as the 100 Hz data above approximately 0.3 Hz. Below this frequency, cospectra remain above zero and ogives do not yet converge to





one, reflecting ongoing flux contributions from turbulent eddies. These results confirm that the 10 Hz GHG analyzer sampling rate, along with any potential dampening by inlet tubing, does not result in significant high-frequency flux loss in our setup.

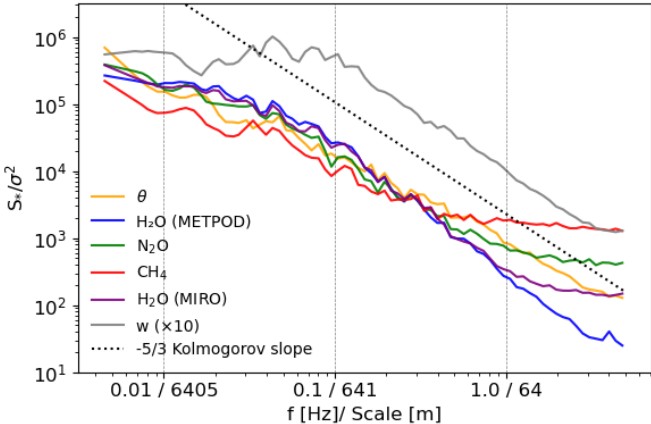

**Figure 5.** Spectral analysis of the flight on 14 June 2023. Frequency multiplied, variance normalized flight average power spectral densities of $N_2O$, $CH_4$, $\theta$, $w$ and $H_2O$ of both instruments, the GHG analyzer and METPOD are shown. $S_{*w}$ is shifted upward by an order of magnitude. For comparison, the -5/3 Kolmogorov slope is depicted as dotted black line. Note the ticks on the abscissa are either frequency or scale size, separated by a slash symbol.

Figure 5 displays the power spectral density for all scalars and $w$. All spectra are normalized by their variance, and for better visibility $S_{*w}$ is shifted upwards by an order of magnitude. The fast spectra of $H_2O$ and $\theta$ from METPOD, as well as of $w$, align well with the theoretical $-5/3$ slope of the inertial subrange, extending to the Nyquist frequency of the GHG analyzer (5 Hz).

Scalars measured by the GHG analyzer level off at the high-frequency end, indicating white noise. For $CH_4$, frequencies above 0.5 Hz cannot be resolved, for $N_2O$ and $H_2O$ the white noise is apparent at frequencies above 1 Hz. Crucially, the noise occurs at frequencies well above the flux-relevant range ($\approx 0.015 - 0.3$ Hz) identified in Figure 4. Therefore, the presence of white noise does not impair flux calculations. The dominant flux-contributing scales lie within the inertial subrange, consistent with turbulence theory and confirming the suitability of our measurements for eddy covariance flux estimation.

## 3.2 Flux uncertainty contributions

EC is subject to uncertainties related to the statistical nature of turbulence, limitations of the measurement setup and uncertainties arising from vertical flux divergence. Individual sources of error and their calculation for the GHGMon flights are listed in Appendix A.

Figure 6 illustrates the relative contributions of different uncertainty sources for both short and long leg lengths. The pie

charts represent averages over the four flights shown in Figure 3 and listed in Table 1. Total uncertainties, expressed as the ratio of average total error to mean flux ($\frac{\overline{\sigma}_{\text{tot.}}}{\overline{F}}$) are 60 % for $N_2O$ and 76 % for $CH_4$ when using short legs ($L =8500$ m). These values decrease to 47 % and 60 %, respectively, for longer legs ($L=15300$ m). In all cases, random error ($\overline{\sigma}_{\text{tot.rand.}}$) is the dominant





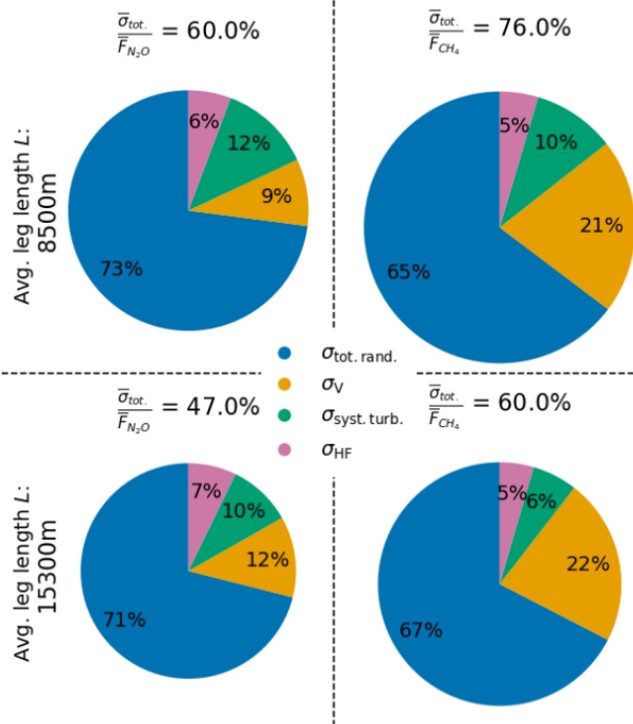

**Figure 6.** Flux error composition shown as average pie charts over the four flights depicted in Figure 3. The left column displays relative uncertainty contributions for N$_2$O fluxes, and the right column for CH$_4$. The top row corresponds to short legs ($L =$8500 m), while the bottom row shows results for long legs ($L =$15300 m).

contributor, accounting for approximately 70 % of the total uncertainty for both trace gases and across leg lengths. This term includes, besides decorrelations caused by natural inhomogeneities, uncertainties related to the limited precisions of the instruments, but also possible artificial decorrelations between vertical wind and trace species, caused by small imperfections

in the water vapor correction. Reducing this component would be feasible through the use of higher-precision instrumentation, both the GHG analyzer and the five-hole probe, as well as by targeting more homogeneous emission sources. Of these two options, the former will be feasible with future progress in GHG analyzer hardware, better spectral fitting algorithms, and improvements of the five-hole probe. The second depends on the emission source itself. While thoughtful selection of leg

positions and lengths to better fit spatial characteristics can slightly reduce uncertainty, this only holds as long as fundamental EC requirements are not violated (e.g., by excluding large-scale contributions through overly short legs). Source heterogeneity imposes a natural lower limit on the lowest achievable uncertainty in airborne EC flux measurements. The systematic turbulence error, $\overline{\sigma}_{\mathrm{syst.turb.}}$, scales inversely with $L$, and thus decreases for longer averaging periods. In contrast, the relative contributions of the uncertainties caused by the vertical flux divergence correction, $\overline{\sigma}_{\mathrm{V}}$, and by the high frequency loss, $\overline{\sigma}_{\mathrm{HF}}$, increase with $L$,

as they are independent of $L$. $\overline{\sigma}_{\mathrm{HF}}$ could be reduced by increasing the sample turnover rate in the analyzer's cavity. Balancing



high spatial resolution against low flux uncertainties requires consideration of the overarching research objectives. Besides deploying more precise instruments, additional uncertainty reduction could be achieved through repeated and more extensive vertical flux divergence experiments, including more repetitions across a greater number of altitudes. Continuous wavelet transform analysis holds the potential to overcome the trade-off between spatial resolution and uncertainty (Vaughan et al., 2021; Wolfe et al., 2018; Metzger et al., 2013), and will be applied in the future to study the GHGMon data.

In addition to assessing uncertainty, it is essential to determine the limits of detection (LOD) to characterize the smallest detectable fluxes for individual flight legs. This allows to evaluate whether a measured flux is distinguishable from noise. Following Rannik et al. (2016), the noise of single-leg fluxes was determined based on precisions of both the scalar concentration $c$ and vertical wind speed $w$ (see Appendix A), and thus provides an estimate of the flux LOD. Table 2 summarizes the calculated LODs for $N_2O$, $CH_4$, $\lambda E$ and $H$ across the four flights, along with their averages.

**Table 2.** Flux-LODs for $N_2O$, $CH_4$, $\lambda E$ and $H$. For each research flight, single leg LODs were averaged. The last column is the average of all four flights.

| - | 14 June | 21 June a | 21 June b | 26 June | Average |
|---|---|---|---|---|---|
| $\mathrm{LOD}_{\overline{F}_{N_2O}}$ in $\mu\mathrm{g\,m^{-2}\,s^{-1}}$ | 0.042 | 0.028 | 0.036 | 0.044 | 0.037 |
| $\mathrm{LOD}_{\overline{F}_{CH_4}}$ in $\mu\mathrm{g\,m^{-2}\,s^{-1}}$ | 0.16 | 0.11 | 0.14 | 0.17 | 0.14 |
| $\mathrm{LOD}_{\lambda E}$ in $\mathrm{W\,m^{-2}}$ | 1.9 | 1.3 | 1.7 | 1.9 | 1.7 |
| $\mathrm{LOD}_{H}$ in $\mathrm{W\,m^{-2}}$ | 6.9 | 4.4 | 5.9 | 6.9 | 6.0 |

Since LODs are calculated for individual flight legs, they are not directly comparable to values from other measurement sites, especially with different meteorological conditions. Furthermore, the LOD reduces by averaging repeated overflights of the same leg (Langford et al., 2015). Our setup achieves a relatively low LOD for $CH_4$ fluxes of $0.14\,\mu\mathrm{g\,m^{-2}\,s^{-1}}$. This is comparable to the range reported by Wiekenkamp et al. (2025) $(0.1-0.14\,\mu\mathrm{g\,m^{-2}\,s^{-1}})$, and notably better than the average LOD of $0.66\,\mu\mathrm{g\,m^{-2}\,s^{-1}}$ reported by Pasternak (2023). To our knowledge, the only other airborne EC $N_2O$ flux study, estimated an LOD of $0.1\,\mu\mathrm{g\,m^{-2}\,s^{-1}}$ (Wilkerson et al., 2019). Our system achieves an even lower value of $0.037\,\mu\mathrm{g\,m^{-2}\,s^{-1}}$.

## 4   Agricultural GHG emissions in Friesland: Emissions strengths and spatiotemporal variability in June 2023

In this section we first demonstrate the ability of our approach to deliver spatially resolved $N_2O$ fluxes, as derived along single flight legs. We then compare our flux results for repeated flight legs at similar locations to demonstrate that derived emissions are consistent within uncertainties and expected variability, respectively. We further discuss the temporal variability of both $N_2O$ and $CH_4$ emissions by presenting the flux results from airborne measurements conducted on different days before and after a rain event. Finally, we compare the $N_2O$ emission rates with regional emissions reported in inventories for Friesland and with emissions observed in other agricultural areas worldwide. Please note that the focus in this section is on $N_2O$ with only some references to $CH_4$ results, motivated by the lack of airborne studies on agricultural $N_2O$ emissions.



## 4.1 Spatial variability of N$_2$O fluxes

Figure 7 shows N$_2$O fluxes ($L = 7600$ m, 10 s leg to leg shift) for the four Friesland flights (see Table 1) plotted as time-series and corresponding locations plotted on a map.

**Figure 7.** N$_2$O surface fluxes of four research flights over Friesland. The left panels shows fluxes from 14 June, the two panels in the middle from the forenoon and the afternoon flight on 21 June, and the right panel from 26 June. The upper row shows the time-series of measured N$_2$O fluxes, the lower row subplots show the spatial distribution of N$_2$O fluxes on maps. Fluxes are color coded, flight averaged mean fluxes are indicated by the black dashed lines and flight averaged LODs are marked as gray shaped areas. Fluxes were calculated using a leg length of $L = 7600$ m, a leg to leg shift of 10 s and are overlaid on Google Earth imagery (Image © Landsat/Copernicus, Maxar Technologies).

The time-series of N$_2$O fluxes show little leg-to-leg variations on 14 June and 26 June, with some decent emission peaks in between, rarely exceeding 0.25 µg m$^{-2}$ s$^{-1}$. In contrast, fluxes measured during the two flights on 21 June show some sections of small fluxes and several sections with very high emissions up to around 1 µg m$^{-2}$ s$^{-1}$. Both, flight mean fluxes (dashed black lines) as well as leg-to-leg variability (spread of the fluxes) are significantly higher on 21 June than on the other two



days. The lower part of Figure 7 reveals that the highest fluxes on 21 June were detected in relatively confined regions in the southern/central part of the flight patterns. Repeated legs over the same ground scene, but flown during different times of the flights, and even at different times of the day (forenoon flight 21a, afternoon flight 21b), yield similar emission rates. This reproducibility indicates spatial and temporal coherence of the flux signals. The small-scale variability of $N_2O$ fluxes, from close to zero in some parts of the flight area up to emissions of around $1 \, \mu g \, m^{-2} \, s^{-1}$ (in the central part of the patterns) demonstrates the ability of our airborne EC setup to detect and hence, spatially resolve small-scale (i.e. $1-10 \, km$) emission hotspots within a relatively homogeneous landscape. Differences in observed $N_2O$ emission fluxes at flight level may be caused by either rapid changes of $N_2O$ emission rates at the surface or by small changes in footprint areas due to stronger turbulence, slightly higher wind speed (plus $\approx 1.4 \, m/s$) and slightly more westerly wind direction (plus $\approx 14 \, °$) during the afternoon flight on 21 June. In general, our findings of high spatial variability including observations of confined $N_2O$ emission hotspots agree well with results reported by Dacic et al. (2024). As mentioned before, a detailed flux footprint calculation is part of an upcoming study, but we note that the meteorological conditions (especially wind direction and speed) were relatively constant throughout the day, justifying our conclusion of spatially consistent fluxes at a qualitative level.

## 4.2 Temporal variability of $N_2O$ and $CH_4$ fluxes

Figure 8 shows flight-averaged $N_2O$ and $CH_4$ fluxes for the four GHGMon flights analyzed in this study. Measured $N_2O$ fluxes were notably higher during the two flights on 21 June 2023, as already shown in Figure 7, compared to lower values on 14 and 26 June. The difference between the highest and lowest $N_2O$ fluxes spans a factor of approximately three. Peak $N_2O$ emissions coincide with elevated flight averaged $\lambda E$ of close to $1000 \, W \, m^{-2}$. In contrast, $CH_4$ fluxes showed less variation between flights ($1.21 \check{} 1.86 \, \mu g \, m^{-2} \, s^{-1}$) and did not peak on 21 June. There was no significant precipitation throughout the first half of June 2023, implying a low volumetric soil water content, as evident by ERA5 reanalysis data (Copernicus Climate Change Service, 2023; Hersbach et al., 2020). A frontal system accompanied by thunderstorms passed over Friesland on the evening of 20 June 2023, leading to intense precipitation and elevated soil moisture. During the subsequent research flights on 21 June 2023, hot (25 °C) and sunny weather with scattered cumulus clouds prevailed. These conditions, high surface temperatures, strong solar heating, and wet soils, favor elevated $\lambda E$. Furthermore, it is known that intense precipitation events and dry-wet transitions are typical hot-moment scenarios, leading to peak $N_2O$ emissions (Anthony and Silver, 2021; Butterbach-Bahl et al., 2013). Our observations of peak $N_2O$ emissions during the post-precipitation period strongly support this phenomenon. Eckl et al. (2021) made similar observations, with largest $N_2O$ fluxes measured after a flooding event in the U.S. Midwest. Both regions, the U.S. Midwest and Friesland, are characterized by intense agricultural activities, including the use of manure and synthetic fertilizer (Miller et al., 2012; Van Der Heide et al., 2011), which are key contributors to $N_2O$ emissions (Stehfest and Bouwman, 2006). Therefore, our findings of enhanced $N_2O$ fluxes during dry-wet transitions in agriculturally dominated regions like Friesland are also consistent with prior studies. Fluxes measured on 26 June are comparable to those during the low soil moisture conditions on 14 June, even though soil water content was as high as during 21 June with the hot-moment emissions (see Figure 8). This finding gives additional evidence, that $N_2O$ emissions peak during changing soil water content, not necessarily during periods of high soil water content (Barrat et al., 2021; Harris et al., 2021; Butterbach-Bahl et al., 2013).





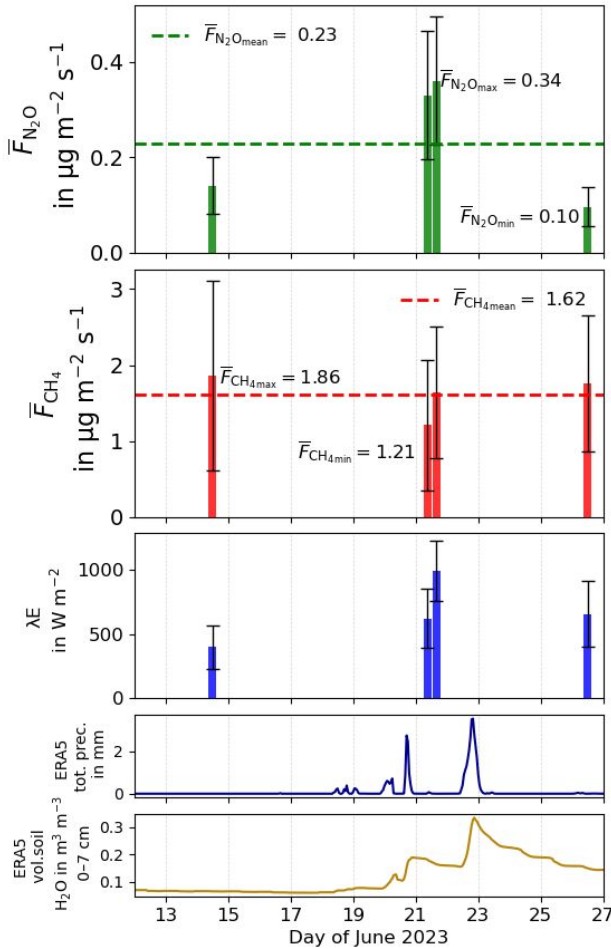

**Figure 8.** Flight average fluxes of $N_2O$ and $CH_4$ for four flights conducted during the GHGMon campaign. Green bars represent $F_{N_2O}$, red bars $F_{CH_4}$ and thin black bars the associated total error margin. The blue bars in the third row are latent heat fluxes, $\lambda E$. The lowermost subplots indicate the total precipitation and volumetric soil water content from the surface down to -7 cm, for the region between 53.3°N, 5.2°W and 52.2°N, 6.2°W, both inferred from ERA5 reanalysis data.

In contrast, the relatively stable $CH_4$ fluxes indicate a source less sensitive to short-term meteorological changes, likely enteric fermentation from ruminants. This interpretation is supported by isotopic measurements from JAS flask samples collected during the four flights, which revealed a $\delta^{13}C(CH_4)$ isotopic signature of -61.2±2.3 ‰, closely matching the -60.8±0.2 ‰ signature reported by Röckmann et al. (2016) for agricultural, ruminant-dominated emissions at the Cabauw tall tower near

Utrecht, Netherlands. BU emission estimates of Emissieregistratie also show, that the majority of $CH_4$ emissions in Friesland is produced by ruminants. A deeper interpretation of the flux observations and a spatially resolved comparison with available BU inventories will follow in future work. The results from the four example flights demonstrate that our EC data can be used



to enhance understanding and quantification of agricultural GHG emissions across multiple scales: from regional inventory comparison like Eckl et al. (2021); van der Laan et al. (2009) or Fu et al. (2017), and evaluation of process-based models e.g.
Ma et al. (2024) or Zhang et al. (2017), to the identification of small-scale emission hotspots (see Figur 7). Furthermore, our results of spatially confined emissions hotspots, as well as hot-moment emissions of $N_2O$, with differences by a factor of three within days on the regional-scale, demonstrate the importance of spatially and temporally comprehensive observations. Future work will include a more detailed investigation of external emission drivers, ideally combined with supporting information like activity data or ground-based observations, as done in Van Der Poel et al. (2025) for $CO_2$.

**4.3 Comparison of $N_2O$ emissions in Friesland with inventories and other agricultural regions worldwide**

Here we compare our calculated regional, total $N_2O$ emissions both with results from other studies worldwide and with two emission inventories.

Across the four research flights, we observed a mean $N_2O$ flux of $0.23\,\mathrm{\mu g\,m^{-2}\,s^{-1}}$, with peak fluxes of $0.34\,\mathrm{\mu g\,m^{-2}\,s^{-1}}$ on June 21. These peak values are at the upper limit of maximum fluxes reported in comparable studies of agricultural $N_2O$
emissions during the spring-summer period (see Table 3).

van der Laan et al. (2009) reported $Rn^{222}$ tracer-based $N_2O$ fluxes of $0.174\,\mathrm{\mu g\,m^{-2}\,s^{-1}}$ in the northern Netherlands, the same general region as our study. This value is close to our mean and they also observed daily flux variability of similar magnitude to ours. Kroon et al. (2010) measured EC fluxes on the field-scale of a dairy farming area near Reeuwijk, Netherlands, with peak emissions $> 0.3\,\mathrm{\mu g\,m^{-2}\,s^{-1}}$ during the growing season, similar to our findings. They inferred a multi-annual mean
flux (2006$-$2008) of $0.07\,\mathrm{\mu g\,m^{-2}\,s^{-1}}$, which is comparable to our fluxes before and after the hot-moment emissions on 21 June. Compared to the findings of Eckl et al. (2021), who reported a mean $N_2O$ flux of $0.047\,\mathrm{\mu g\,m^{-2}\,s^{-1}}$ and a maximum of $0.079\,\mathrm{\mu g\,m^{-2}\,s^{-1}}$ for the U.S. Midwest, our values are considerably higher, although both study regions are similarly dominated by agriculture. Fu et al. (2017) recorded a mean flux of $0.193\,\mathrm{\mu g\,m^{-2}\,s^{-1}}$ in the U.S. Corn Belt in June, using a combination of tower and flask measurements with model simulations. Their results closely match our mean flux of $0.23\,\mathrm{\mu g\,m^{-2}\,s^{-1}}$),
and their deviation from the results from Eckl et al. (2021) for the same region highlight the high spatio-temporal variability of agricultural $N_2O$ emissions. Dacic et al. (2024) conducted airborne $N_2O$ measurements on three subsequent days in May 2022 across a region near Des Moines, Iowa, within the U.S. cornbelt. They inferred fluxes using a Bayesian inversion modeling framework and found peak emissions of $0.38\,\mathrm{\mu g\,m^{-2}\,s^{-1}}$ on one day and a mean flux of $0.29\,\mathrm{\mu g\,m^{-2}\,s^{-1}}$ across all three days, very similar to our findings for the four Friesland research flights. Based on tall tower EC measurements in the
western Pannonian Basin, Hungary,Haszpra et al. (2018) reported a mean flux of $0.010\,\mathrm{\mu g\,m^{-2}\,s^{-1}}$ and a maximum flux of $0.093\,\mathrm{\mu g\,m^{-2}\,s^{-1}}$ between March and August. Their lower mean may reflect regional differences in agricultural practices or the inclusion of months with low emissions. Still, they also observed peak fluxes up to $0.93\,\mathrm{\mu g\,m^{-2}\,s^{-1}}$ following heavy rain, consistent with the pattern of our findings. Murphy et al. (2022) observed a mean of $0.123\,\mathrm{\mu g\,m^{-2}\,s^{-1}}$ and a maximum of $0.436\,\mathrm{\mu g\,m^{-2}\,s^{-1}}$ from near-surface EC in Ireland after fertilizer application and rainfall — values comparable to ours.
Table 3 also includes inventory-based fluxes. From the EDGAR v8.0 database (European Commission. Joint Research Centre. and IEA., 2024), a mean $N_2O$ flux of $0.015\,\mathrm{\mu g\,m^{-2}\,s^{-1}}$ was inferred for June 2023 over the region spanning $53.3-52.2\,°N$





**Table 3.** Comparison of our $N_2O$ fluxes with observed $N_2O$ fluxes in agricultural regions worldwide and with EDGAR v8.0 and the Dutch national inventory (Emissieregistratie). All numbers in units of $\mu g \, m^{-2} \, s^{-1}$.

| Work | $\overline{F}_{N_2O_{mean}}$ | $\overline{F}_{N_2O_{max}}$ | Method | Location | Season |
|---|---|---|---|---|---|
| This Work | 0.23 | 0.34 | Airborne EC | Friesland, NL | June |
| van der Laan et al. (2009) | - | 0.174 | Atmospheric obs. + reference tracer | Lutjewad, NL | June |
| Kroon et al. (2010) | 0.07* | > 0.3[†] | EC + empirical regression | Reeuwijk, NL | *yearly average (2006-2008) [†]growing season |
| Eckl et al. (2021) | 0.047 | 0.079 | Airborne obs. + model simul. | Midwest of US | June/July |
| Fu et al. (2017) | 0.193 | - | Atmospheric obs. + model simul. | Cornbelt, US | June |
| Dacic et al. (2024) | 0.29 | 0.38 | Airborne obs. + model simul. | Iowa, US | May |
| Haszpra et al. (2018) | 0.010 | 0.093 | Tall tower EC | Western Pannonian Basin, HU | March - August |
| Murphy et al. (2022) | 0.123 | 0.436 | EC + flux chamber | Johnstown Castle, IE | June - September |
| EDGAR v8.0 | 0.015 | | | (53.3°N, 5.2°E) - (52.2°N, 6.2°E) | June (2023) |
| Emissieregistratie | 0.014 | | | Friesland | annual average (2022) |

and $5.2 - 6.2$ °E. This area was chosen to include Friesland and the area covered by our flights. Although no footprint analysis has been performed yet, the EDGAR value is approximately one order of magnitude lower than our airborne EC-derived flux, even for the days at which we measured our smallest fluxes. This discrepancy may partly arise from mismatched footprint areas,

but also highlights the weak seasonal variability in EDGAR data. The difference between minimum and maximum monthly means for 2023 is only about 10 %, which contrasts with findings in literature that report pronounced seasonal peaks during the growing season (Eckl et al., 2021; Butterbach-Bahl et al., 2013)). Similarly, the Dutch national inventory (Emissieregistratie) reports a mean flux of 0.014 $\mu g \, m^{-2} \, s^{-1}$ for Friesland, representing the annual average for 2022. This value closely matches the EDGAR estimate and is again significantly lower than our measurements. Besides the potential footprint mismatch, the use

of annual averages in Emissieregistratie may under-represent elevated emissions on shorter time-scales, like during the growing season. While this comparison to inventory emissions is subject to several limitations, the discrepancies strongly suggest a too



weakly pronounced annual cycle and, hence, an underestimation of growing-season $N_2O$ emissions in the inventories. A more detailed and robust comparison, including footprint analysis, will be the focus of future work.

## 5   Summary and Conclusions

This paper introduces a novel airborne eddy covariance system for the regional-scale quantification of $N_2O$ and $CH_4$ fluxes, while giving additional high spatial resolution. The system consists of an aircraft-adapted commercial QCL-based absorption spectrometer, which delivers 10 Hz data of $N_2O$ and $CH_4$, and the METPOD meteorology measurements, including fast (up to 100 Hz) vertical wind data, hence being able to determine vertical turbulent GHG fluxes. We successfully conducted measurements over Friesland, the Netherlands, in June 2023, using the DLR research aircraft Cessna Grand Caravan. We

characterize the MIRO MGA$^3$ GHG analyzer and its airborne modifications, describe an in-flight calibration routine, and apply a suite of data quality control criteria to comply with the basic assumptions of EC. Emission fluxes were controlled for well-developed turbulence, and associated uncertainties are in line with those reported in airborne studies characterizing less complex sources from the fossil fuel or waste sector, with relative errors around 30 % under favorable conditions, and larger than 100 % for single legs with low fluxes. Spectral analysis of $N_2O$ and $CH_4$ fluxes confirm consistency with turbulence

theory. Power spectra of both GHGs comply to the -5/3 slope in the inertial subrange and show white noise only beyond 1 Hz, indicating negligible high-frequency losses. To investigate possible non-turbulent, mesoscale contributions on the low-frequency end of the spectrum, a high-pass filtering approach based on $w$ spectra and the isotropy of turbulence was applied. The resulting emission fluxes, are not impacted by mesoscale contributions. Average detection limits are 0.037 $\mu g\, m^{-2}\, s^{-1}$ for $N_2O$ and 0.14 $\mu g\, m^{-2}\, s^{-1}$ for $CH_4$, demonstrating the high sensitivity of our setup, particularly for airborne $N_2O$ flux

measurements. Emission fluxes derived from four research flights conducted over Friesland revealed relatively constant $CH_4$ emissions ($\pm 25$ % around 1.62 $\mu g\, m^{-2}\, s^{-1}$). In contrast, $N_2O$ emissions showed strong temporal coupling with soil moisture dynamics and were characterized by spatially consistent emission hotspot regions. The observed hot-moment fluxes of $N_2O$ are more than three times larger than fluxes during constant soil moisture and align with findings from previous studies on agricultural $N_2O$ emissions. Mean $N_2O$ fluxes (0.23 $\mu g\, m^{-2}\, s^{-1}$) are among the highest reported globally for growing-season

agricultural emissions, and preliminary comparisons with EDGAR v.8.0 and the Dutch national inventory Emissieregistratie indicate substantial under-representation of growing-season $N_2O$ emissions and the lack of an appropriate annual cycle. While limitations in spatial footprints and temporal averaging must still be considered, which will be addressed in an upcoming study, these discrepancies underscore the relevance of this sector for GHG emissions in the Netherlands and the need for further measurements. Overall, our airborne EC method offers a robust tool for quantification of agricultural emissions and

enhancing understanding of the processes driving them. It represents a significant step forward in top-down quantification of anthropogenic sources on the regional-scale, extending beyond approaches that focus primarily on fossil fuel and waste sectors. Moreover, the approach can be extended to constrain emissions from key natural sources, including $N_2O$ release from natural soils and oceans and $CH_4$ emissions from wetlands, contributing to a more complete understanding of the global greenhouse gas budget.



*Data availability.*   The GHGMon data are available from the authors upon request

## Appendix A:  Airborne EC uncertainty analysis

Various sources of uncertainty arise during EC emission estimation. Some uncertainties stem from imperfections in the measurement setup, while others result from the statistical nature of the measured variables, such as turbulent transport processes in the PBL. Additionally, violations of the core assumptions of EC theory further increase uncertainties in the calculated fluxes.

These errors are classified as either systematic or random. Systematic errors lead to directed false estimates (biases), such as the underestimation of true fluxes for very short flight legs, as large-scale eddy contributions are excluded. In contrast, random uncertainties reduce the significance of a flux value, for example, due to a limited number of individual samples during the flux calculation period. Lenschow et al. (1994) estimated the relative difference of the ensemble flux and the true flux as follows:

$$\frac{\sigma_{\text{syst.turb.}}}{F} \leq 2.2 \cdot \frac{z_{\text{meas}} \cdot z_i^{0.5}}{L} \tag{A1}$$

This is an upper limit approximation of the systematic error due to the finite number of samples (i.e. missing large-scale contributions) and is hence independent of the measured species. The systematic error can be reduced by either increasing $L$ (which in turn limits spatial resolution), or by flying lower, i.e. reducing $z_{\text{meas}}$. The determination of $z_i$ from vertical profiles depends on the variable used—such as $H_2O$, $\theta$, wind vector, or GHGs—and can vary between them. Since Equation A1 defines a maximum of the systematic turbulence uncertainty, the variable yielding the largest $z_i$ was selected.

Relative random errors due to turbulence and the associated scattering of flux measurements around the true flux can be calculated via (Lenschow et al., 1994):

$$\frac{\sigma_{\text{rand.turb.}}}{F} \leq 1.75 \cdot \frac{z_{\text{meas}}^{0.25}}{(L \cdot z_i)^{0.5}} \tag{A2}$$

Additional random errors originate from limited instrument precision and contain white noise contributions from the GHG analyzer and the five-hole probe. Following Rannik et al. (2016), they can be expressed as:

$$\sigma_{\text{instr.noise}} = \frac{(\sigma_c^2 \tilde{\sigma}_w^2 + \sigma_w^2 \tilde{\sigma}_c^2)^{0.5}}{N} \tag{A3}$$

Here, $\sigma_w^2$ and $\sigma_c^2$ are the variances of the vertical wind and the scalar measurements, and $\tilde{\sigma}_w$ and $\tilde{\sigma}_c$ are the precisions of the vertical wind and scalar measurements, respectively. $N$ is the number of samples taken in the flux segment. Equation A3 was used to define flux-related LODs, representing the minimum flux that the EC system can reliably distinguish from zero.

A mathematical approach constraining both random turbulence and random noise errors is outlined in Finkelstein and Sims

(2001). This method uses co- and autocovariance terms at different lag times $\tau$, where $\tau$ should be greater than the integral time scale, to directly derive the flux variance from the time-series measurements:

$$\sigma_{\text{tot.rand.}} = \left( \frac{\sum\limits_{i=-\tau}^{\tau} \overline{c'c_i'}\,\overline{w'w_i'} + \overline{w_i'c'}\,\overline{c_i'w'}}{N} \right)^{0.5} \tag{A4}$$



In this equation, $c'$ and $c_i'$ represent the turbulent (fluctuating) components of the scalar concentration, with $c_i'$ being shifted by $i$ time steps relative to $c'$. The same notation applies to $w$. In the following analysis, Equation A4 is used to determine the total random uncertainties in the flux calculations. This choice is made, because Equation A2 provides only an upper limit and does not account for instrumental influences, while Equation A3 neglects errors arising from turbulence sampling.

Special flight patterns were designed to empirically capture this vertical flux divergence during GHGMon. To minimize the influence of spatial and temporal variability in the fluxes, we flew stationary legs stacked in the vertical and one immediately after the other in the early afternoon. We flew at 300 m, 390 m, 450 m, and 690 m altitude, while $z_i$, inferred from profiling, was 890 m. Each level was flown twice, with $L \approx 50$ km. We calculated single-level fluxes, and extrapolated the surface fluxes from their altitude dependence using orthogonal distance regression (ODR). Based on Wolfe et al. (2018) a flux divergence correction factor $V$, depending on $z_{\mathrm{meas}}$ was calculated:

$$V = \frac{t}{m \cdot z_{\mathrm{meas}} + t} \tag{A5}$$

Here, $m$ is the slope of the linear regression and $t$ the intercept at ground level. Errors in both, $m$ and $t$ add uncertainties to the extrapolated surface flux:

$$\sigma_{\mathrm{V}} = \sqrt{\left(\frac{\partial V}{\partial t}\sigma_t\right)^2 + \left(\frac{\partial V}{\partial m}\sigma_m\right)^2} \tag{A6}$$

The relative uncertainty of the vertical flux divergence correction is $\sigma_{\mathrm{V}}/V$.

The total uncertainty for a flux segment is determined by propagating the errors of all individual error terms, resulting in:

$$\frac{\sigma_{\mathrm{tot}}}{F_{\mathrm{surf}}} = \sqrt{\left(\frac{\sigma_{\mathrm{tot.rand}}}{F}\right)^2 + (\sigma_{\mathrm{tot.syst.}})^2 + \left(\frac{\sigma_{\mathrm{V}}}{V}\right)^2} \tag{A7}$$

Here, $\sigma_{\mathrm{tot.syst.}}$ is the overall systematic error, calculated as:

$$\sigma_{\mathrm{tot.syst.}} = \sqrt{\left(\frac{\sigma_{\mathrm{syst.turb.}}}{F}\right)^2 + \sigma_{\mathrm{HF}}^2} \tag{A8}$$

Applying error propagation to systematic uncertainties is challenging because they can be directional, and in some cases, both the magnitude and sign of the bias is known, allowing for potential correction. However, since $\sigma_{\mathrm{syst.turb.}}$ is estimated as an upper limit, applying a correction would likely overcompensate. The high-frequency loss term, $\sigma_{\mathrm{HF}}$ (from Equation 6), is based on the assumption of cospectral similarity, a modeled cospectral shape, and the determination of $f_{\mathrm{m}}$ and $\tau$, making it uncertain. Additionally, for our fast GHG analyzer, $\sigma_{\mathrm{HF}}$ is relatively small compared to other error terms, meaning its impact on the flux is negligible. For these reasons, both systematic error terms are treated the same as random ones and were not used to correct fluxes.

## Appendix B: Comparison to flask measurements

Figure B1 compares our high-frequency measurements, corrected for water vapor and calibrated in flight, with discrete flask samples from JAS for four GHGMon flights. For both $N_2O$ and $CH_4$, the correlation coefficients between the GHG analyzer



measurements and JAS flasks are high ($R^2$ values of 0.97 for $N_2O$, 0.92 for $CH_4$), demonstrating the reliability of the water vapor correction and calibration procedure. We quantify the analyzer's accuracy using the mean absolute differences $\varepsilon$ between the two instruments, taking JAS as the reference. This yields $\varepsilon_{N_2O} = 0.36$ ppb and $\varepsilon_{CH_4} = 7$ ppb. Minor differences between

the instruments are expected due to their different integration times: the GHG analyzer has a response time of 0.16 s, whereas sampling a single JAS flask takes several tens of seconds. For the comparison, we smoothened GHG analyzer data and observed no systematic difference between the two instruments under varying environmental conditions, such as during vertical profiles, aircraft maneuvers, or across different ambient temperatures.

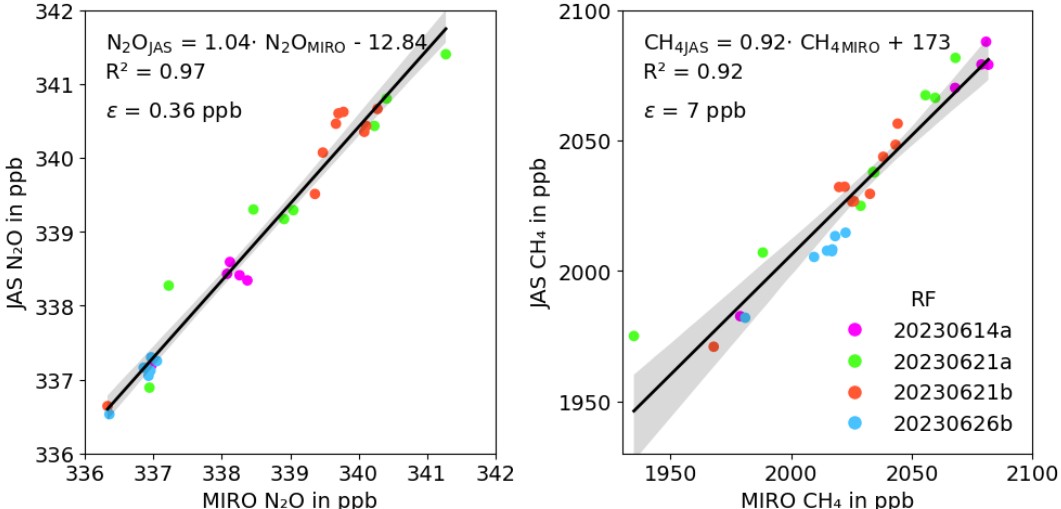

**Figure B1.** $N_2O$ and $CH_4$ measurements from the GHG analyzer and JAS flasks of four flights during GHGMon. Each research flight is marked with a distinct color. Despite some outliers, the overall comparison between the two instruments shows strong agreement, with $R^2$ values of 0.97 for $N_2O$ and 0.92 for $CH_4$, and regression slopes close to one. The mean absolute deviation, $\varepsilon$, quantifies the difference between the two instruments. Assuming the JAS flasks to represent the true values, $\varepsilon$ indicates the campaign-averaged accuracy of the MIRO GHG measurements.

## Appendix C: Keeling analysis of Friesland $\delta^{13}C(CH_4)$

*Author contributions.* PW wrote the paper, did the analysis and prepared the figures. PW, ME, LK, KG, CM, RH and AR did the flight planning. PW, ME, LK, MG, AR and CM prepared the instruments. PW, ME, LK, KG and MG conducted the measurements. TR, RH and HC supported the campaign planning with their local knowledge. AR led the aircraft campaign, CK led the DLR project GHGMon. All authors contributed to the manuscript and the discussion.



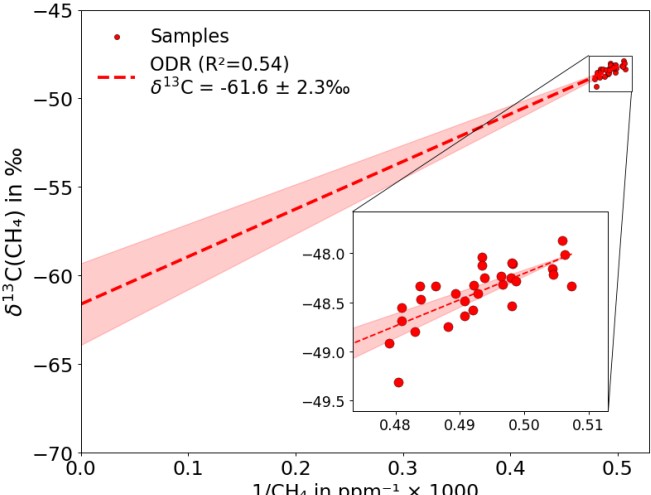

**Figure C1.** Keeling plot of the JAS samples taken during the four flights over Friesland. No significant flight to flight difference was observable.

*Competing interests.* Thomas Röckmann and Huilin Chen are members of the editorial board of AMT.The remaining authors declare that
they have no conflict of interest

*Acknowledgements.* The authors thank DLR-FX for the campaign cooperation, especially the pilots Thomas van Marwick, Miguel Pereda Albarran and Götz Hieber, in-flight technicians David Woudsma and Christoph Grad, the sensor group of Christian Mallaun and Vladyslav Nenakhov, project manager Oliver Paxa and flight operations officer Andrea Haushold and Felix Betsche.



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
