# Peer review of "Quantifying agricultural N2O and CH4 emissions in the Netherlands using an airborne eddy covariance system"

_EGUsphere, 2025_

## Author Comment (AC1)

**Answer to referee comment 1 for "Quantifying agricultural N₂O and CH₄ emissions in the Netherlands using an airborne eddy covariance system"**

We thank reviewer 1 for the evaluation of our manuscript, especially for pinpointing sections with possibility to improve clarity. Below we list our answers to the comments and corresponding revisions in the manuscript. Referee comments are written in normal font, our answers in italic font.

The present study "Quantifying agricultural N2O and CH4 emissions in the Netherlands using an airborne eddy covariance system" describes in detail a measurement system and airborne eddy covariance method for quantifying CH4 and N2O fluxes over an agricultural region. The study is generally clearly presented and extremely detailed in methodology. Novel findings of N2O emissions from agriculture and comparison to inventories are also presented. The authors thoroughly consider uncertainties and potential biases in the measurements. I recommend publication following minor revisions.

Line 32: I believe the current IPCC recommendation for the GWP100 of biogenic CH4 is 27.

Thank you for this correction, 27 is the most recent value IPCC assessed for GWP100 of biogenic CH4. We changed the corresponding sentence in the manuscript.

General comments/questions, largely for the purposes of improved clarity:

1: I assume that the choice of 90 s was at least partially made to capture all of the eddy scales based on ogives, analysis of the cospectral power, and/or integral timescale. I don't believe an explanation of this was explicitly given in the manuscript. It would be helpful to see in the text a description of what factors went into to choosing the 90 s windows.

The choice of an appropriate window length for flux calculation is an important aspect of our work, and not straight forward, especially not for airborne applications. Hence, we derived and evaluated the window length: Assumption "zero" was based on typical ground-based EC flux calculation intervals. Those are typically 30 min in length. We have translated those 30 min timescales into a length-scale by multiplying it with typical horizontal wind speeds (3-6 m/s), yielding 5-11 km. With our average aircraft speed of 62 m/s, we need to fly 90 – 180 s to capture the same (5-11 km) length scale. Hence, 90 s was our first minimum window length to start with. Then, as presented in Section 3.1, we used cospectral analysis and the ogive method to test whether our 90 s window length is sufficient or not. As demonstrated by the ogives for all cases, even for those with long flux legs (large window sizes) we don't see significant flux contributions at the low-frequency end of the spectrum (frequencies below 0.014Hz). Therefore, we can conclude, that our window length is long enough to capture all relevant flux carrying eddies.

We adapted corresponding paragraphs in Sections 2.1.2 and 3.1 to improve the description of our approach of choosing the 90 s window length.

2: I found it generally confusing what is meant by leg, vs Flight leg, vs flux segment. It would be useful if clear definitions were explicitly given and/or more consistency in the language were used. e.g. Do these terms always refer to the flight leg, or sometimes to the 90 s intervals as it seemed?

Thank you for this hint. To avoid confusion by inconsistent terminology, we changed our terms to "flight path", when ever talking about a certain part of the whole flight pattern (e.g. between two turns). When referring to a certain distance, for which a single flux value is calculated, we introduced the term "flux leg". Flux leg replaces the former terms of leg, flux segment and further. The temporal pendant to flux leg is the size of the moving window.

3. The authors mention that spatial homogeneity is required for eddy covariance, but the flux variations over a leg seem to indicate non-homogeneity. I would assume the condition of homogeneity is only necessary over the 90 s windows use for the flux calculations. Does the overlapping windows further loosen this condition? Some discussion of this in the text would be useful.

We indeed found a significant spatial variability of fluxes across the target area (Hotspots vs. areas with low fluxes), hence heterogeneity in the emission strength. The eddy covariance method formally requires homogeneity of the underlying flux field, but as you point out, only across the window for which the flux is calculated. This implies stationary turbulence, a constant footprint area and a spatially uniform surface flux within the flux leg. In practice, for airborne fluxes this condition is rarely met for entire target areas, since flight tracks inevitably cross heterogeneous landscapes. The moving-window approach allows us to resolve spatial variability of fluxes along the flight track, but it also implies a relaxation of the EC assumption of surface homogeneity. Each window provides a flux estimate under the assumption of local homogeneity within the corresponding footprint, while the overall heterogeneity of the landscape is captured by the variability across consecutive windows. Variability between consecutive windows can then be interpreted as spatial heterogeneity in the flux field, and is not a violation of the EC assumption of homogeneity.

We added sort of this discussion of flux heterogeneity and its relation to the moving window approach partly to the description of the moving window technique in the manuscript Section 2.1.2:

"Short flux legs allow for higher spatial resolution and are more likely to be located within a homogeneous source area, but potentially miss flux contributions from large eddies (see Section 2.1.3) and are associated with higher flux uncertainties because of smaller sample sizes (see Section 3.2). Overly long flux legs are prone to violations of source homogeneity and do not offer fine spatial resolution."

And partly to section 4.1, where we discuss spatial variability of fluxes:

"Repeated flux legs over the same ground scene, but flown during different times of the flights, and even at different times of the day (morning flight 21a and afternoon flight 21b), yield similar emission rates. This reproducibility indicates spatial coherence and temporal consistency of the flux signals, hence stationarity. We observe smooth transitions in fluxes of single flux legs between regions of low fluxes and regions of high fluxes and since consecutive flux legs have a large spatial overlap with corresponding similar footprint, local homogeneity within the flux legs can be assumed. Thus, the variability of  $N_2O$  fluxes across the target area, from close to zero in some parts of the area up to emissions of around 1  $\mu$ g m-2 s-1 (in the central part of the patterns) demonstrates the ability of our airborne EC setup to detect and hence, spatially resolve small-scale (i.e. 1–10 km) emission hotspots within a relatively homogeneous landscape."

4. Are the LODs calculated for each 90 s segment (i.e. N represents the number of observations per 90 s flux interval)? If so, are these simply averaged over the flight? If, rather, N is the entire leg, wouldn't this calculation of LOD underestimate the ability of the instrumentation to

distinguished spatially-resolved fluxes? In general I think more clarification is needed to contextualize the LODs reported.

Yes, LODs are calculated for each window (90 s segment). We use the moving window approach to achieve high spatial resolution by inspecting fluxes of consecutive windows, hence we need to define the LOD of each single window to assess whether the corresponding flux value can reliably be differentiated from zero.

We clarified in the manuscript that LODs were calculated for each single window and complemented the manuscript by the following paragraph, with the aim to provide a better insight to the meaning of flux LODs for the reader: Beyond using LODs to distinguish between pure random noise and true flux, they can be used to compare different EC systems based on their smallest detectable flux. However, this comparison is somewhat limited, since the LOD depends on instrumental noise, turbulence conditions, and window length, hence varies across different flux legs, as turbulence intensity and scalar variance can change, which directly propagate into the LOD calculation. It is therefore not only system specific, but also dependent on turbulence and the true flux field. Thus, single flux leg LODs are no constants but should be interpreted as context-dependent thresholds of confidence. By averaging LODs over multiple flux legs and across different flights, we can achieve a better comparability to other airborne EC systems.

Specifically, we added the following paragraph:

"Beyond using LODs to distinguish between pure random noise and true flux, they can be used to compare different EC systems based on their smallest detectable flux. However, this comparison is somewhat limited, since the LOD depends on instrumental noise, turbulence conditions, and window length, hence varies across different flux legs, as turbulence intensity and scalar variance can change, which directly propagate into the LOD calculation. Furthermore, the LOD reduces by averaging repeated overflights of the same flux leg (Langford et al., 2015). LODs are therefore not only system specific, but also dependent on turbulence and the true flux field. Thus, single flux leg LODs are no constants but should be interpreted as context-dependent thresholds of confidence. By averaging LODs over multiple flux legs and across different flights, we can achieve a better comparability to other airborne EC systems. Our setup achieves a relatively low averaged LOD for CH4 fluxes of 0.14 μg m-2 s-1. This is comparable to the range reported by Wiekenkamp et al. (2025) (0.1–0.14 μg m-2 s-1), and notably better than the average LOD of 0.66 μg m-2 s-1 reported by Pasternak (2023). To our knowledge, the only other airborne EC N2O flux study, estimated an LOD of 0.1 μg m-2 s-1 (Wilkerson et al., 2019). Our system achieves an lower value of 0.037 μg m-2 s-1."

5. It would be helpful to have a figure on the flux divergence calculation in the Appendix.

We put the following figure into the Appendix and used it to explain how we assessed the vertical flux divergence correction, by adding the text paragraph beyond.

"Figure A1 shows measured averaged fluxes of H,  $\lambda E$ , CH4 and N2O, normalized by their extrapolated surface flux values, for the different altitude levels probed during the vertical flux divergence experiment. Similar slopes of normalized fluxes of all scalars indicate a gradual decrease of measured fluxes from the surface upwards.  $z_i$  was inferred to be 890 m amsl. on this day (21 June 2023) by vertical profiles."

6. Authors mention recent studies utilizing the continuous wavelet transform method, which is often thought of as better for obtaining higher spatial resolution. Is there a reason that the authors used the moving window method instead?

Continuous wavelet transform (CWT) is sometimes applied in airborne flux measurements and offers advantages in cases of strong spatial heterogeneity or non-stationary conditions with e.g. intermittent turbulence. Its main benefit is the high resolution in both frequency and time domain. We used classical EC partly because we compensate for the superiority of CWT by using appropriate moving windows: They allow for high spatial resolution, limit the assumption of stationarity to only the short windows, and with cospectral analysis, we demonstrate that we don't suffer from low-frequency loss.

On the other hand, despite the advantages, the use of CWT requires careful consideration of several factors, including the low-frequency cutoff for cospectrum integration. This low-frequency cut-off (using Fourier transformation) is also part of the discussion in our work, and by default, with CWT one integrates contributions from all scales, conceivably non-turbulent large-scale contributions, which must not to be interpreted as instantaneous surface flux (Li et al., 2023). Secondly, the choice of an appropriate mother wavelet (e.g., Morlet or Mexican hat) is not always clear and can lead to different quality of the flux calculation under different conditions (Mi et al., 2005, Schaller et al., 2017). Third, the validity of flux estimates near the edges of flux legs (cone of influence) has to be considered. The overall aim of this study is to demonstrate that agricultural greenhouse gas fluxes can be reliably quantified with the new airborne system. For  $N_2O$  fluxes, this is a new approach to our knowledge. Therefore, we have chosen classical eddy covariance as an appropriate first step, given its long-standing establishment and acceptance in the flux measurement community, before applying the more advanced CWT.

We have added a short statement (to not confuse the reader with methodological details), why we have chosen classical EC in the paragraph where we mentioned CWT:

"Continuous wavelet transform analysis holds the potential to overcome the trade-off between spatial resolution and uncertainty (Vaughan et al., 2021; Wolfe et al., 2018; Metzger et al., 2013), but its application is outside the scope of this study, as its implementation requires careful methodological choices and interpretation, while the classical EC approach provides a well-established and reliable framework for this first demonstration of our system."

**References**

Mi, X., Ren, H., Ouyang, Z., Wei, W., and Ma, K.: The use of the Mexican Hat and the Morlet wavelets for detection of ecological patterns, Plant Ecology, 179, 1–19, https://doi.org/10.1007/s11258-004-5089-4, 2005.

Schaller, C., Göckede, M., and Foken, T.: Flux calculation of short turbulent events – comparison of three methods, Atmospheric Measurement Techniques, 10, 869–880, <a href="https://doi.org/10.5194/amt-10-869-2017">https://doi.org/10.5194/amt-10-869-2017</a>, 2017.

Li, Y., Wu, Y., Tang, J., Zhu, P., Gao, Z., and Yang, Y.: Quantitative Evaluation of Wavelet Analysis Method for Turbulent Flux Calculation of Non-Stationary Series, Geophysical Research Letters, 50, e2022GL101 591, https://doi.org/10.1029/2022GL101591, 2023.

---

## Author Comment (AC2)

**Answer to referee comment 2 for "Quantifying agricultural N₂O and CH₄ emissions in the Netherlands using an airborne eddy covariance system"**

We thank the reviewer 2 for the recommendations for improvement of our manuscript. Below we list our answers to the comments and corresponding revisions in the manuscript. Referee comments are written in normal font, our answers in italic font.

The authors describe high-speed airborne in situ measurements of N2O and CH4 using a new commercial spectrometer. These measurements are used to calculate vertical fluxes of N2O and CH4 via eddy covariance, and results are described for four flights during the GHGMon campaign in 2023. Results show a persistent level of CH4 flux consistent with ruminant emissions and increased N2O flux within a day of a significant rain event, with evidence that N2O flux increases is related to an increase but not amount of soil moisture. I find this work to be novel, well written, and worthy of publication in Atmospheric Chemistry and Physics. I recommend publication after the authors address a few minor comments:

Section 2: Placing the instrument and campaign description subsections before the details of the eddy covariance analysis may be a better way to organize the paper. I found myself jumping down to the instrument & campaign section and finally just read it totally before going back to the EC section.

The positioning of subsections in the methodology part of our manuscript (Eddy covariance theory vs. instrument and campaign description) was indeed under debate among the participating authors. Since we don't assume Eddy covariance to be very well known within the expected audience, we decided in the end to start with the principles of Eddy covariance. We are aware that this may limit the smoothness of readability somewhat, but this would likely be the same, when discussing e.g. measurement strategies and designated eddy covariance flight patterns (in the campaign section) before introducing the eddy covariance methodology itself.

As a compromise and to improve the reading flow, we expanded the short summary/introduction of the overall methodology Section (Section 2), to give all relevant information on the instrumentation to the reader which is necessary to go through the Eddy covariance Section more smoothly.

Line 135: You denote stationarity, horizontal homogeneity, and well-developed turbulence as core EC assumptions, and I think you demonstrate each of these at different points, but the explicit terminology doesn't come back. I'm not sure the exact place(s) where it would be appropriate, but it would be good somewhere in the manuscript to denote exactly where you are justifying these assumptions (e.g. sect 3.1 for stationarity, see comment below).

Thank you for this hint, we added discussion to revisit the underlying concepts and explicitly state how they are fulfilled in the corresponding Sections.

• You are right, we demonstrate (quasi-) stationarity using the cospectral analysis in conjunction with our method to estimate the cut-off frequency between turbulent and non-turbulent, larger scale motions. We don't find significant low-frequency, large-scale contributions, from e.g. mesoscale motions, which would have led to a break-down of the stationarity assumption.

- Horizontal homogeneity is now discussed in more detail in the context of the moving window approach, where only local homogeneity is required within each single window. By comparing fluxes of consecutive windows, possible spatial heterogeneity of the flux field can be determined with this method, instead of leading to a violation of the homogeneity assumption.
- We used the convective velocity scale w\* to define a threshold for well-developed turbulence of individual flux legs. By inspecting variance spectra of vertical wind and scalars we find a pronounced inertial subrange spanning a broad range of frequencies following the expected -5/3 law, indicating that turbulence is well developed with energy cascading down from the energy containing range to the dissipation range. Under these conditions, the covariance calculation provide a meaningful representation of the flux.

Specifically, we have included (inter alia) the following paragraphs to

**Section 4.1:**

"Repeated flux legs over the same ground scene, but flown during different times of the flights, and even at different times of the day (morning flight 21a and afternoon flight 21b), yield similar emission rates. This reproducibility indicates spatial coherence and temporal consistency of the flux signals, hence stationarity. We observe smooth transitions in fluxes of single flux legs between regions of low fluxes and regions of high fluxes and since consecutive flux legs have a large spatial overlap with corresponding similar footprint, local homogeneity within the flux legs can be assumed. Thus, the variability of  $N_2O$  fluxes across the target area, from close to zero in some parts of the area up to emissions of around 1  $\mu$ g m-2 s-1 (in the central part of the patterns) demonstrates the ability of our airborne EC setup to detect and hence, spatially resolve small-scale (i.e. 1–10 km) emission hotspots within a relatively homogeneous landscape."

**Section 3.1:**

"This marginal shift of dominating eddy scale sizes after applying the high-pass filter indicates that contributions from non-turbulent, large-scale motions are negligible and do not distort stationarity during the time of the flight, which is a core assumption of EC. Nonstationarity would have led to incomplete ogives, with overshoots or oscillations."

"The dominant flux-contributing scales lie within the inertial subrange, which in this case is well pronounced over a broad range of frequencies for all scalars and w, demonstrating well-developed turbulence and hence consistency with EC theory."

Line 160: Upcoming manuscript that by now may be citable? Or an upcoming experiment? It may suffice to simply state that you used Equation 4 and deeper footprint analysis is deemed outside the scope of this manuscript.

We changed our comments on the footprint calculation as suggested. We are working on more advanced footprint analysis currently, for future comparison of the GHGMon data with emissions inventories at high spatial resolution. But as you point out, those activities are out of the scope of this manuscript, which aims for the introduction of the airborne eddy covariance system and the demonstration of its suitability to study agricultural GHG emissions.

Line 167: This line about the timing is important to include, but was confusing placed here in the middle of a discussion of empty data treatment (made me ask if you were interpolating GPS time synchronization data).

We adapted the corresponding description of clock synchronization for improved clarity. We were not interpolating GPS time stamps, but due to small divergence between GHG and METPOD clock, we had to interpolate single GHG data values a few times per flight.

Line 240: Should "no" be "minimal" or "no significant"? ogives give no further contributions only when they reach the highest/lowest frequency at 1 or 0...so maybe once contributions reach N%?

That is true, our initial statement that "ogives don't show further contributions" implies a misleading impression of exactness. We changed this accordingly and used the formulation of "no further significant contributions".

Line 301: What WMO scale is your dataset traceable to, presumably X2006A?

Yes, it is X2006A. We included this information in the manuscript.

Line 310: Have you examined whether there is any artifact from aircraft motion in the measurements? If there is, then preferentially calibrating in turns could bias the dataset.

We did visual inspection of data quality during aircraft maneuvers, including turns, climbs, descents and at higher flight levels with lower ambient pressure. We could not find a dependency (bias) of any of the measured GHG concentrations related to these maneuvers, nor were calibration gas measurements in turns or during climbs and descents of less quality (like enhanced noise (spread)).

Line 311: Have you evaluated whether transferring standard gases affects the in-flight concentrations? Usually when standards are filled, there is some amount of settling time.

We are aware that the transfer of reference gases can lead to a reduced quality of the calibration and has to be done with caution. During GHGMon, the refilling of gases was done thoroughly by flushing the pressure reducers of the involved cylinders and the transfer-pipes at least five times generously with calibration gas. Furthermore, the same large gas cylinder was used for the replenishment of the small cylinders during the whole period, hence no residual gas contamination was possible. In the comparison of the GHG analyzers concentration measurement with the flask samples of JAS (Appendix), we don't see significant flight to flight discrepancies, which would elucidate a possible contamination of calibration gas due to gas transferring. We have therefore high confidence that our calibration routine does not suffer from the gas transfer.

Line 315: Did you evaluate whether a calibration slope correction is needed, even if you could not perform this in the air (e.g. in the lab, on the ground with other calibration standards).

We acknowledge that a slope correction (with a two-point calibration) leads to a higher accuracy in most cases, but as mentioned, we were not able to apply a two-point calibration during the GHGMon campaign. We indeed made experiments with multiple reference gas bottles of different  $N_2O$  concentrations in the laboratory and derived corrections. We found that at least between 300ppb and 400ppb mole fractions of  $N_2O$ , a linear correction (slope correction) fits the observations versus reference standard best. Those slope corrections were relatively constant over repeated experiments with a couple of weeks in between. We observed variability of slopes between minimum 1.004 and maximum 1.028 (without systematic trend between minimum and maximum slopes). Changes in the slope are a consequence of changing sensitivity, which indeed can affect eddy covariance fluxes.

A slope correction propagates directly into the flux calculation according to:

$$F_{true} = \overline{w'c'_{true}} = \overline{w'(m \cdot c_{meas} + t)'} = m \overline{w'c_{meas}'} = m F_{meas}$$

With subscript "true" denoting real flux or concentration (with slope correction), "meas" denoting raw flux and concentration (without slope correction), m the slope of the linear fit, and t the intercept. t does not contribute to the covariance term, since it is only a constant added to the concentration. In the case of slope m=1, the calibration has no impact on the fluxes ( $F_{true}=F_{meas}$ ). If  $m\ne 1$ , the relative flux error due to a missing slope correction can directly be inferred from the slope. Our laboratory experiments revealed slopes slightly deviating from 1 (between 1.004 and 1.028), therefore, and under the assumption that the sensitivity of the GHG analyzer would have shown similar behavior during the GHGMon campaign as it did in the laboratory, the associated additional relative flux error would only have been 0.4-2.8 %. This assumption is confirmed by the comparison of the GHG analyzer with the flask samples (JAS), where a slope of 1.04 was found, which also only slightly deviates from 1, but additionally includes effects of limited accuracy from JAS (see Appendix B).

Due to the expected small flux error contribution of a missing slope correction compared to other flux error terms, and due to the fact that we could not perform a slope correction during the campaign, hence could not quantify this error term, we decided to not include it in our calculations. Since we acknowledge that a slope correction would slightly improve our flux estimates, and since this information might be of use for similar approaches, we included our statement in the Airborne EC uncertainty analysis Section in Appendix A:

"In the case of  $N_2O$ , we furthermore note, that our limitation to a simple offset calibration will introduce an additional flux error, directly proportional to the difference of the slope from one. During repeated laboratory experiments with different reference gases, we found slopes only slightly deviating from one (maximum 1.028), hence the corresponding errors are expected to be very small compared to other flux error terms (<3-4%). Therefore, because of other flux error terms being an upper limit approximation, and because we could not determine this uncertainty contribution quantitatively, we decided to not include it in our uncertainty assessment."

Line 322: This seems important to evaluate the measurements, there should at a minimum be a reference to this other publication, at least a conference proceeding/presentation.

The water vapor correction is definitely important to consider for the flux calculations. There will be a separate publication on this topic in the near future, but by the time of the final submission of this manuscript, this separate publication is not citable. We inserted instead a reference to a conference presentation, where we addressed the water vapor correction.

Line 334: "measured with a time resolution" -> "reported at", 100 Hz is not the sampling rate of any of the METPOD suite, just the sampling rate.

This was a mistake by the authors. 100 Hz is just the rate at which meteorological and wind measurements are reported. This was corrected in the manuscript accordingly, for example:

"The vertical wind component is reported with 100 Hz and an uncertainty below 0.2 m s-1. Spectral analysis revealed, that up to 10 Hz wind measurements, no noise contributions or system-specific resonance frequencies affect the data quality."

Line 407: If I understand correctly, this comparison is a pseudo-test of stationarity, but I don't see you mention that directly.

Yes, it is, we adapted the paragraph accordingly. See also answer to comment in Line 135 above.

Figure 4: It would be nicer to the eye for the x scales to be the same for each panel.

Thank you for pointing this out. We changed the x axis tick arrangement to allow a direct comparison of the different panels.

Line 421: 100 Hz is the data rate but not the time response of METPOD (see comment above). I couldn't find in the reference the exact time response for the METPOD TAT measurement (I only saw a power spectrum for the wind components), but I've never seen a PT100 TAT sensor with a faster response than 7-8 Hz. Acknowledged that this does not affect the point you are making with the converging ogives by 0.3 Hz.

We apologize for the same erroneous phrasing similar to the comment to Line 334. We removed the statement of a 100 Hz measurement frequency. Instead, we use the phrasing "fast" and refer to Mallaun et al. (2015), where it is shown, that the wind measurement (including vertical wind speed) is at least reliable up to undisturbed measurement frequencies of 10 Hz. We have proven in our work, that this is clearly fast enough for the computation of the eddy covariance fluxes. Regarding the TAT measurement, we agree with the referee that 100 Hz is not the time response of the sensor, but in analogy to the wind measurements, we can state that based on our spectral analysis, the TAT measurement has no white noise contribution down to the Nyquist frequency of 5 Hz, hence at least has an effective measurement frequency of 5 Hz - fast enough for the GHGMon fluxes.

Line 446: Can you tell if the GHG sensor or wind sensor are the limiting noise source, or if they both contribute similarly?

Thanks for this question. We investigated the contributions to the instrumental noise error upon your question, by separately computing the terms including the precision of the GHG analyzer and the vertical wind measurement, since there is no interplay between those variables in the instrumental noise error term  $\sigma_{instr.noise}$  (Appendix Equation A3). We found that for N2O, the GHG analyzers contribution to the instrumental noise error is 85% compared to 15% of the vertical wind measurement (average over the flights presented in our work). For CH4, the GHG analyzers contribution is 69% on average, and the one of the vertical wind measurements 31%. We

included this information in the uncertainty discussion (Section 3.2), since it might be of interest when thinking about technical optimization of our or comparable systems.

"We notice that instrumental noise contributions from the GHG analyzer are approximately five to six times larger than those from the vertical wind measurements in the case of  $N_2O$ , and approximately two to three times larger for the  $CH_4$  flux error (averaged across all four flights)."

Figure 7 caption: forenoon -> morning

The word "forenoon" was replaced with "morning" in all parts of the manuscript.

Line 530: Figur -> Figure

Corrected!

**References**

Mallaun, C., Giez, A., and Baumann, R.: Calibration of 3-D wind measurements on a single-engine research aircraft, Atmospheric Measurement Techniques, 8, 3177–3196, https://doi.org/10.5194/amt-8-3177-2015, 2015.